# Oral Cholecalciferol Supplementation in Sahara Black People with Chronic Kidney Disease Modulates Cytokine Storm, Oxidative Stress Damage and Athero-Thromboembolic Risk

**DOI:** 10.3390/nu14112285

**Published:** 2022-05-29

**Authors:** Houda Zoubiri, Amina Tahar, Samir AitAbderrhmane, Messaoud Saidani, Elhadj-Ahmed Koceir

**Affiliations:** 1Laboratory of Biology and Organisms Physiology, Team of Bioenergetics and Intermediary Metabolism Nutrition and Dietetics in Human Pathologies Post Graduate School, University of Sciences and Technology Houari Boumediene, El Alia, Bab Ezzouar, Algiers 16123, Algeria; hd.zoubiri@gmail.com (H.Z.); taharamina87@gmail.com (A.T.); 2Biology and Physiology Laboratory, Ecole Nationale Supérieure de Kouba, Algiers 16308, Algeria; 3Diabetology Unit, Seghir Nekkache Hospital, Ain Naâdja, Algiers 16105, Algeria; saitabderrahmane@yahoo.fr; 4Clinical Nephrology Exploration Dialysis and Kidney Transplantation Unit, University Hospital Center of Beni Messous, Algiers 16014, Algeria; saidanime@yahoo.fr

**Keywords:** chronic kidney disease, racial and ethnicity, black South Sahara (SS) residents, white SS residents, Vitamin D supplementation, proinflammatory cytokines, oxidative stress, atherothrombogenic risk

## Abstract

The 25-hydroxyvitamin D_3_ (25OHD_3_) deficiency in chronic kidney disease (CKD) is associated with immune system dysfunction (pro-inflammatory cytokines storm) through macrophages renal infiltration, oxidative stress (OxS) damage and athero-thromboembolic risk. Conversely, cholecalciferol supplementation (25OHD-S) prevents kidney fibrosis by inhibition of vascular calcification and nephrotic apoptosis (nephrons reduction). The objective of this study was to investigate the pleiotropic effects of 25OHD-S on immunomodulation, antioxidant status and in protecting against thromboembolic events in deficiency CKD Black and White individuals living in the Southern Sahara (SS). The oral 25OHD-S was evaluated in 60,000 IU/month/36 weeks versus in 2000 IU/day/24 weeks in Black (*n* = 156) and White (*n* = 150). Total serum vitamin D was determined by liquid chromatography-tandem mass spectrometry. All biomarkers of pro-inflammatory cytokines (PIC) were assessed by ELISA tests. OxS markers were assessed by Randox kits. Homocysteine and lipoproteine (a) were evaluated by biochemical methods as biomarkers of atherothromboembolic risk. All statistical analyses were performed with Student’s *t*-test and one-way ANOVA. The Pearson test was used to calculate the correlation coefficient. The means will be significantly different at a level of *p* value < 0.05. Multiple logistic regressions were performed using Epi-info and Statview software. Vitamin D deficiency alters the PIC profile, OxS damage and atherothrombogenic biomarkers in both SS groups in the same manner; however, these disorders are more acute in Black compared to White SS individuals. The results showed that the serum 25OHD_3_ concentrations became normal (>75 nmol/L or >30 ng/mL) in the two groups. We have shown that the dose and duration of 25OHD-S treatment are not similar in Black SS residents compared to White SS subjects, whilst the same inhabit the south Sahara environment. It appears that a high dose intermittent over a long period (D60: 36 weeks) was more efficient in Black people; while a lower dose for a short time is sufficient (D2: 24 weeks) in their White counterparts. The oral 25OHD-S attenuates PIC overproduction and OxS damage, but does not reduce athero-thromboembolic risk, particularly in Black SS residents.

## 1. Introduction

In the southern Sahara (SS) populations, chronic kidney disease (CKD) is estimated at more than 5.8% (3.5 million cases); including Black and White people. In our study, the SS population is heterogeneous in skin pigmentation, race and ethnicity categories including White and Black residents such as Arab, Berbers with Amazigh expression language (Touaregs, Mozabites) and Haratins (slave origin) [1]. Residents are characterized by essential hypertension, which represents the major risk factor to renal failure with 50.2% of prevalence [2]. In 2021, global epidemiological data estimate CKD in more than 10% of the world’s population, averaging 697 million cases in all CKD stages [3]. CKD is usually defined from a KDIGO (Kidney Disease Improving Global Outcome) by estimated Glomerular Filtration Rate (eGFR) <60 mL/min per 1.73 m^2^ [4]. KDIGO proposes that the calculation of eGFR equation in adults’ CKD should take into account the racial-ethnic factor [5]. Furthermore, the 25-OHD deficiency is associated with the depletion of FGF-23 (Fibroblast Growth Factor-23), which leads to hyperphosphatemia [6]. In contrast, an accumulation of FGF-23 is a major factor of renal α-hydroxylase inhibition that leads to depletion of 1,25-dihydroxyvitamin D [1,25(OH)2D] in CKD people [7]. Some studies reveal that Black people have higher serum phosphate and lower urinary phosphate excretion despite higher levels of phosphaturic hormones, FGF-23 and PTH [8]. Previous studies show the contribution of calcium, phosphorus, and 25-OHD to the excessive severity of secondary hyperparathyroidism in African Americans with CKD and may contribute to the increased bone mass in Black individuals [9].

Among several factors that contribute to renal deterioration in CKD, ongoing low-grade inflammation is dominant. It is characterized throughout by acute–phase proteins, such as C-reactive protein (CRP), fibrinogen, and albumin [10]. The CKD evolution accentuates the inflammatory state, which is mediated by renal immune cells infiltration and significant proinflammatory cytokines secretion such as Tumor Necrosis Factor-α (TNF-α) and interleukins [11]. Besides, CKD inflammation state is recurrently related to thrombogenic process and myocardial infarction, particularly, the increase homocysteine serum levels [12] and recently linked lipoprotein (a) [13]. In addition, oxidative stress is involved in the progression of renal injury, pathogenesis of atherosclerosis, and exacerbation of disease burden in CKD patients.

Additionally, oxidative stress is strongly associated with CKD inflammation by releasing a massive amount of ROS (reactive oxygen species) [14], which lead to accumulation of extracellular oxidative proteins, podocyte damage, mesangial expansion, renal hypertrophy, tubulointerstitial fibrosis, and glomerulosclerosis. Thus, oxidative stress and inflammation further contribute to cardiovascular events and mortality in CKD end-stage renal failure and renal dialysis [15]. The renal dysfunction is protected from the detrimental effects of ROS, mainly by the antioxidant enzymatic system (AES), which includes SOD (superoxide dismutase), GPx (glutathione peroxidase) and catalase [16]. The ROS neutralization is conducted primarily by AES through antioxidant trace elements (ATE) integrated as AES cofactors, such as selenium (Se), copper (Cu), manganese (Mn) and zinc (Zn). Recently, it has been shown that selenium (Se) plays a crucial role through immunomodulatory and anti-inflammatory properties [17]. 25OH D (vitamin D_3_) plays an essential role between immune system and kidney, as immune cells express 1-alpha-hydroxylase to regulate local 1,25-dihydroxyvitamin D_3_ or calcitriol (active form) concentration [18]. Concomitantly, 25OH D plays a crucial function between oxidative stress (OxS) and kidney [19]. The calcitriol effects are exerted via VDR receptor (Vitamin D Receptor) present on T2 lymphocytes. Calcitriol inhibits proinflammatory cytokines, such as interleukins (IL) 1 and IL-6; and increased anti-inflammatory cytokines IL-4 and IL-10 [20]. Vitamin D_3_ deficiency in Black people is more pronounced than in the White population, and may contribute to cardiovascular mortality, including myocardial infarction such as coronary heart disease and heart failure [21]. Some studies have shown the South of Northern Africa, such as Algerian Sahara, as a region with a high prevalence of hypovitaminosis D, and thus requiring higher doses of 2000 IU/day; while in Northern Africa, a dose of 600 IU/day is sufficient to reach a desirable 25OHD level ≥30 ng/mL [22]. Several data showed that the active vitamin D supplementation exerts a beneficial renal protection effect in CKD patients by anti-inflammatory and antioxidant action [23]. A meta-analysis providing several data showed that the active vitamin D or analogues exerts a beneficial renal protection effect in CKD patients by reducing proteinuria and provide anti-inflammatory action [24]. However, NKF-KDOQI studies are controversial about vitamin D supplementation and not the established consensus that defines the optimal level of vitamin D treatment to improve the GFR [25].

In this study, we have tested the supplementation dose–response of 25-hydroyxvitamin D (Cholecalciferol) in continuous low multi-doses (2000 IU/day) versus an intermittent high dose (60,000 IU/month), and to determine the optimal supplementation period (24 weeks versus 36 weeks) in CKD 3 adults’ Black versus White vitamin D deficiency participants from the same region of south-eastern Algeria. According to the pleiotropic Vitamin D3 effects, the objective of this investigation is to know whether the Vitamin D3 supplementation could modulate the following targets: (i) change of serum Vitamin D3 levels; (ii) attenuate pro-inflammatory cytokines production (iii); delete the oxidative stress damage; (iv) and avoid atherogenicity risk. To our knowledge, this study is the first investigation on the according racial/ethnic disparities in the Algerian Saharan population.

## 2. Materials and Methods

### 2.1. Informed Consent Statement

This renal study protocol (Algiers Ethnic-Renal Study) was approved by the ECAMPH (Ethics Committee of Algerian Ministry of Public Health) and conformed to the principles outlined in the declaration of Helsinki (http://www.wma.net, accessed on 1 March 2006). Primary care physician approval after written informed consent was required for all participants. Ethical approval code: The permits and ethical rules have been achieved according to the Executive Decree no. 10–90 (10 March 2010) completing the Executive Decree no. 04–82 (18 March 2004) of the Algerian Government, establishing the terms and approval modalities.

### 2.2. Participants and Study Design

This study was a randomized, multicenter, controlled trial of vitamin D_3_ supplementation in community-based SS Algerian regions (Laghouat, Tamanrasset, Bechar, Adrar, Tindouf, In Salah, Illizi) in CKD peoples not requiring renal dialysis. In the two groups, White (W) and Black (B) participants were recruited into the study in 2015 to 2020; they are 40 to 60 years old. The starting cohort was made up of 760 people who agreed to participate in this study. We proceeded to divide the cohort randomization (Figure 1) into 4 stratified groups depending on the dose of 25-hydroyxvitamin D (Cholecalciferol) to supplement. The dosage schedule of Cholecalciferol (D) supplementation is as follows: The two groups (B versus W) receive two doses of Cholecalciferol treatment (2000 IU/mL capsules every day versus 60,000 IU/mL capsules every month) with two durations (24 weeks versus 36 weeks); however, the 2 groups are divided into 2 subgroups as follows: -Group D2B: 2000 IU/day/24 weeks (*n* = 78);-Group D60B: 60,000 IU /month/36 weeks (*n* = 72);-Group D2W: 2000 IU/day/24 weeks (*n* = 77);-Group D60W: 60,000 IU /month/36 weeks (*n* = 79).

Cholecalciferol was dosed at 2000 IU comes from the Italfarmaco S.A. laboratories, Industrial de Alcobendas, Madrid, Spain. Cholecalciferol dosed at 60,000 IU comes from the Sanofi laboratories Product leaflet of DePURA, India. Regarding 60,000 IU vitamin D_3_ supplementation, Black and White SS CKD participants were supplemented in the beginning with 60,000 IU weekly for 4 weeks, then 60,000 IU for 24 weeks and 36 weeks monthly. All clinical exploration participants have been examined by the same physician. Participants with severe hypocalcemia (<2.02 mmol/L) and hyperphosphatemia (>1.55 mmol/L) were supplemented systematically with 500 mg of calcium carbonate per day in tablet form (Calcidia, Laboratoire Bayer Healthcare, Lyon Cedex 09, France). Demographic, clinical and laboratory data were retrieved from the electronic database of the Diabetology and Nephrology exploration Unit, respectively, in Seghir Nekkache Hospital Center, Ain Nadjaa and in Issad Hassani Hospital Center, Beni Messous of Algiers.

The feeding pattern of the participants is estimated by frequency questionnaire and 24-hour recalls established by the TAHINA in Maghreb region (Transition and Health Impact in North Africa) epidemiological survey [26], which includes 139 questions on 124 foods. The dietary intake, such as vitamin D, calcium, protein, sodium and drinking water have been evaluated from the CIQUAL table [27].

The physical activity level was assessed using the self-administered International Physical Activity Questionnaire (IPAQ) [28]. The participants were classified into four categories: (i) no regular physical activity with a sedentary lifestyle; (ii) minimal physical activity (<75 min/week); (iii) insufficient physical activity (75 min < 150 min/week); and (iv) sufficient physical activity (150 min/week). In addition to physical activity itself, the questionnaire covers four areas: work-related, transportation, housework/gardening, and leisure-time activity.

### 2.3. Clinical Vitamin D3 Supplementation Protocol

In this investigation, we adopted the NKF clinical practice guidelines, which suggest that conventional standard values for 25(OH) D are as follows: (i) vitamin D acute deficiency: <10 ng/mL (<24 nmol/L); (ii) vitamin D mild deficiency: 10–16 ng/mL (24–40 nmol/L); and vitamin D insufficiency: 16–30 ng/mL (40–75 nmol/L) [29]. Recent studies recommend vitamin D supplementation of 50,000 IU/month to prevent osteoporosis. Our study integrates these recommendations [30]; however, it should be noted that the dose to prescribe during vitamin deficiency is still a matter of debate [31].

### 2.4. Chronic Kidney Disease Screening

In the present study, chronic kidney disease (CKD) was explored followed criteria guidelines by clinical practice for the diagnosis, evaluation and prevention CKD treatment according the KDIGO [32]. The CKD stage was evaluated by eGFR using the chronic CKD-EPI equation formula (Chronic Kidney Disease–EPIdemiology collaboration) [33] on the basis of serum creatinine level and expressed for race/ethnicity (black versus white) and sex. The CKD-EPI equation, expressed as a single equation, is GFR = 141 × min (Scr/κ, 1)^α^ × max (Scr/κ, 1)^−1.209^ × 0.993^Age^ × 1.018 (if female) or 1.159 (if black), where Scr is serum creatinine, κ is 0.7 for females and 0.9 for males, α is −0.329 for females and −0.411 for males, in indicates the minimum of Scr/κ or 1, and max indicates the maximum of Scr/κ or 1.

### 2.5. Metabolic Syndrome Clusters Screening

CKD-related Metabolic Syndrome (MetS) components have been diagnosed according to a specific study [34]. The MetS was diagnosed according to the NCEP/ATPIII (National Cholesterol Education Program Third Adult Treatment Panel). The MetS was characterized by the presence of three or more abnormal components as follows: (1) abdominal obesity; (2) high plasma triglyceride level; (3) low plasma HDL cholesterol level; (4) high fasting plasma glucose and (5) a blood pressure disturbance. Medical staff collected anthropometric measurements and blood samples. Body mass index (BMI) was calculated using the standard formula: weight (kg)/height m^2^. Waist circumference was measured at the point between the ribs and iliac crest. Insulin resistance was calculated by the homeostasis model assessment insulin resistance (HOMA-IR) method [35], as follows: HOMA index = fasting glucose (mmol/L) × fasting insulin (mU/L)/22.5 (Normal range: 0.744–2.259). Percent body fat (BF%) was calculated using the formula validated by Deurenberg: BF% = (1.2 × body mass index (BMI) + (0.23 × age) − (10.8 × S) − 5.4 (S is the gender correction factor) [36]. The systolic blood pressure (SBP) and diastolic blood pressure (DBP) were measured in the prone position of the two arms, three times and two minutes after five minutes of rest using a validated Omron 705 CP type BP monitor (Omron Healthcare Europe BV, Amsterdam, The Netherlands) [37].

### 2.6. Blood Samples Analysis Methods

All participants were admitted to the hospital at 7 am after 12 h of fasting before medication. Blood samples from participants were centrifuged at 3000 rpm for 10 min. Serum or plasma samples were immediately put on ice and kept frozen at −80 °C until analyses were performed. Fasting plasma glucose (FPG), triglycerides (TG), total cholesterol (TC), high-density lipoprotein cholesterol (HDL-C) were determined by enzymatic methods using an automatic biochemical analyzer for routine analysis medical laboratory (Cobas Integra 400^®^ analyzer Roche Diagnostics, Meylan, France). FPG was analyzed by using enzymatic hexokinase [38] method. Serum TC was analyzed by colorimetric assay with cholesterol oxidase-peroxidase [39] method. Serum TG was analyzed by colorimetric end-point glycerol phosphate oxidase, phenol + aminophenazone [40] method. Serum HDL-C was analyzed by homogenous enzymatic colorimetric [41] method. The low-density lipoprotein cholesterol (LDL-C) was calculated using Friedewald’s formula [LDL-C (mg/dL) = TC − HDL-C − TG/5.0] [42]. Apolipoprotein A1 (Apo A1), Apolipoprotein B100 (Apo B100), and Lipoprotein (a) (Lp (a)) estimation were carried out on Tulip Quantimate turbidimetry and chemistry analyzer by using immunoturbidimetry method [43]. Albuminuria was categorized as microalbuminuria defined as ≥30 µg/mg was assessed by immuno-turbidimetry [44]. The criterion for detecting low-grade inflammation has been evaluated by serum High-sensitive C-reactive protein (Hs-CRP) levels assessed using immunoturbidimetric methods [45]. Serum fibrinogen levels were measured by the Von Clauss chronometric technique [46]. Serum ceruloplasmin was determined by immunoturbidimetric methods [47]. Plasma homocysteine (hCys) was determined by Fluorescence Polarization Immuno-Assays [48]. Plasma insulin was measured using a double-antibody solid phase radio-immunoassay using Human insulin specific RIA kit, EMD Millipore Corporation St. Louis, MO 63103, USA [49]. Plasma TNF-α [50], IL-1β [51], IL-6 [52], IL-12 [53], IL-18 [54] and IL-23 [55] were determined by ELISA method according to the manufacturers’ instructions (Cayman Chemical’s ACETM EIA kit, Biomedical Campus, Cambridge, CB2 OAX, UK).

### 2.7. Serum Oxidative Stress Screening

Serum trace elements (Selenium, Manganese, Copper and Zinc) were measured by flame atomic absorption spectrometry [56] using AA-680 equipment (Int. Mechanical Equipment Company, Houston, TX, USA). The TAS (Total Antioxidant Status) was analyzed according to previous study [57]. Total Superoxide dismutase activity (SOD) [58], glutathione (GSH) [59] and erythrocyte Glutathione peroxidase (GPx) [60] were analyzed by spectrophotometer methods. The results of erythrocyte SOD and erythrocyte GPx were expressed as U/g of Hb. Plasma TBARS (ThioBarbituric Acid Reactive Substances) estimate in terms of MDA (Malondialdehyde) formed and measured, according to the method described by Ghani et al. [61].

### 2.8. Serum 25-OH Vitamin D and 1, 25(OH) 2D Assessment

Vitamin D status was measured as serum 25(OH) D concentrations (including both D2 and D3 isomers). It should be noted that vitamin D is exposed to a double hydroxylation: the first is located in the liver (1D) to form the 25-OH Vitamin D, and the second is located in the kidney (2D) to synthesize 1, 25(OH) 2D. Both forms of vitamin D were performed using a high performance liquid chromatography (HPLC) method coupled with mass spectrometry (MS/MS). This method consists of extraction steps, pre-chromatography and chromatography of 25OHD. A 100 μL aliquot of serum was mixed with acetonitrile and centrifuged for 10 min at 1500 g. Fifty microliters of the extracted serum samples was injected onto a prepared HPLC system that consisted of a Waters 600E pump, a Waters 717 plus auto sampler, and a Waters 996 photodiode array. We separated samples on a C18 column (150 × 4.8 mm, 5 μm, Supelco, Merck KGaA, Saint-Louis, Missouri, USA) with a 20- × 4.8-mm guard column in line. The HPLC method could separate 1 α-hydroxy vitamin D-3, 25(OH) D2, and 25(OH) D3 using isocratic conditions with a mobile phase that consisted of 87% acetonitrile with all components being detected at a wavelength of 265 nm. The sensitivity of the HPLC method was <10 nmol/L and had a linear range to 1000 nmol/L. The HPLC-coupled mass spectrometer (LCMS-2020, SHIMADZU, Kyoto, Japan) was set for chemical ionization at atmospheric pressure. The analytical cycle time was 5.5 min for a linear calibration for both 25OHD3 (3–283 nmol/L) and 25OHD2 (4–277 nmol/L), with an accuracy of 88–118%; and between 90–100%, respectively [62]. The range of the vitamin D assay is 2.4–114 ng/mL or 6–285 nmol/L. Serum 1,25(OH)2D was measured by the method of Reinhardt et al. [63]. The range of the 1, 25(OH) 2D assay is 18–78 pg/mL or 43.2–187 pmol/L.

### 2.9. Statistical Analysis

All statistical analyses were performed with Epi-info version 5 and Statview version 5 (Abacus Concepts, Berkeley, CA, USA). Data (normally distributed) are presented as mean ± standard deviation (SD). Student’s *t*-test and one-way ANOVA were used for the comparison between the CKD Black versus CKD White groups for the same dose of vitamin D supplementation: D2W versus D2B and D60W versus D60B, mainly to analyze the relationship between vitamin D deficiency in CKD Black and White groups. Pearson’s correlation analysis was performed to quantify associations between the each dose effects of vitamin D supplementation and pro-inflammatory cytokines profile, oxidative stress status, atherogenicity biomarkers and MetS clusters. Multiple logistic regressions were used to assess the association between the 25OHD-S effects and the biological parameters in this study with their 95% confidence intervals. This analysis was performed using Epi-info and Statview software(Version 7.2, Microsoft Windows 7, Wake County, NC, USA). The results were considered significant at * *p* < 0.05, very significant at ** *p* < 0.01 or highly significant at *** *p* < 0.001.

## 3. Results

### 3.1. Clinical Cohort Characterization

Clinical characteristics of SS population by sex and race/ethnicity are described in Table 1. Referring to skin color, we have listed more Black SS residents (53.7%) than White SS people (46.3%) without difference of age or sex. In this study, the chronic kidney disease (CKD) significantly affects (*p* < 0.001) many more Black SS people (49%) than White SS people (37%). Another important fact, essential hypertension (74.9%), obesity (86%) and hypocalcemia (60.6%) are the dominant pathologies associated to CKD in the Black SS group versus White SS group (60.5%, 60% and 42.8%, respectively, *p* < 0.001). Hyperuricemia and congestive heart failure are other comorbidities that are associated with CKD in Black SS group, but moderately elevated compared to White SS. However, coronary artery disease, peripheral vascular disease, dyslipidemia and anemia are more common in Whites SS compared to Blacks SS (*p* < 0.01). We did not notice any difference for the smoking current and alcohol use. Medication treatment (current drug use) is the same for Black SS and White SS groups. Surprisingly, Black SS group seem to consume more vitamin D than White SS group, while both groups are deficient in vitamin D. In addition, caloric intake is significantly higher (*p* < 0.001) in Black SS group compared to White SS group. In contrast, calcium intake is significantly important (*p* < 0.001) in the White SS group compared to Black SS group. The Black SS group is mostly a sedentary lifestyle versus White SS group.

### 3.2. Effects of 25OHD-S on the Inflammation Status

#### 3.2.1. Systemic Inflammation

Serum Hs-CRP levels were significantly higher in the Black SS participants (Table 2) compared to White SS counterparts. Pearson’s correlation analysis shows that Serum Hs-CRP levels were positively associated with micro albuminuria levels and negatively correlated to the urinary albumin–creatinine ratio in Black SS group, but not with eGFR (r = +0.78, *p* < 0.001; r = −0.63, *p* < 0.01, respectively). However, fibrinogen levels were not statistically significant between Black SS (Table 2 and Table 3) and White SS counterparts. The data mentioned in Table 2 and Table 3 shows that cholecalciferol supplementation (25OHD-S) had a drastic anti-inflammatory effect on dropping serum Hs-CRP levels and led to a significant reduction compared to the baseline state in both SS groups. At D2-36 weeks, a significant decrease (−40%, *p* < 0.001) was seen in the White SS group before and after vitamin D treatment, whereas a significant decrease (−34%, *p* < 0.01) was observed at D60-36 weeks in Black SS groups. This anti-inflammatory effect was not observed with fibrinogen.

#### 3.2.2. Pro Inflammatory Cytokines Profile

Our study reveals that plasma basal TNF*-α* levels were elevated in both groups, but more increase in Black SS compared to White SS counterparts. Plasma TNF*-α* levels were inversely correlated to 25-OHD deficiency in both groups (r = −0.91, *p* < 0.001; r = −0.67, *p* < 0.01, respectively). In contrast, plasma TNF*-α* levels were positively and significantly correlated to insulin resistance and visceral adiposity expressed by waist circumference in both SS groups, but more markedly in Black participants (r = +0.77, r = +0.86, *p* < 0.001, respectively). This study indicated that 25OHD-S has a significant beneficial effect on decreasing the circulating TNF-α at follow-up compared with baseline SS groups (Figure 2C). Plasma TNF*-α* level declined significantly at D2-36 weeks in the Whites SS group before and after 25OHD-S. Inversely, a significant decrease was been obtained at D60-36 weeks in Black SS groups.

Plasma basal Interleukin-1β (IL-1β) concentrations were significantly higher in Black SS compared to White SS counterparts. Plasma IL-1β levels were inversely correlated to eGFR in Black SS, but not White SS counterparts (r = −0.59, *p* < 0.001; r = −0.82, *p* < 0.001, respectively). Plasma IL-1β levels significantly dropped from baseline to 36 weeks after 25OHD-S in both SS groups (Figure 2A). The results obtained shows that IL-1β level was decrease significantly at D2-36 weeks in the Whites SS group before and after 25OHD-S. Contrariwise, a significant reduce was been obtained at D60-36 weeks in Black SS groups.

Plasma basal Interleukin-6 (IL-6) concentrations were significantly increased in Black SS participants compared to White SS counterparts. Plasma IL-6 levels were positively correlated to BMI and the systolic blood pressure in Black SS but not White SS counterparts (r = +0.37, *p* < 0.02; r = +0.33, *p* < 0.02, respectively). Plasma IL-6 levels were significantly reduced by the 25OHD-S compared to baseline in both SS groups (Figure 2B). Plasma IL-6 level decreased significantly at D2-36 weeks in the Whites SS group before and after vitamin D treatment. Oppositely, a significant reduce was observed at D60-36 weeks in Black SS groups.

Plasma basal Interleukin-12 (IL-12) levels were significantly higher in Black SS participants compared to White SS counterparts. Plasma IL-12 levels were positively correlated to Apo B_100_/Apo A_1_ ratio, but not correlated to plasma homocysteine or plasma Lp (a) in Black SS group compared to White SS group (r = +0.57, *p* < 0.001). Plasma IL-12 concentrations were a slowly mitigated by the 25OHD-S compared to baseline in both SS groups (Figure 3A). Plasma IL-12 level was diminished at D2-36 weeks in the Whites SS group before and after 25OHD-S. Inversely, in the Black SS group, a reduction was noted at D60-36 weeks.

Plasma basal Interleukin-18 (IL-18) levels were significantly lower in Black SS compared to White SS counterparts. We found strongly positive correlation between eGFR depletion and decreased plasma IL-18 levels in Black SS participants compared to White SS counterparts (r = +0.92, *p* < 0.001). Plasma IL-18 levels were inversely correlated with waist circumference (r = −0.49, *p* < 0.001), Homa-IR (r = −0.38, *p* < 0.001), fasting insulin (r = −0.27, *p* < 0.001) and systolic blood pressure (r = −0.54, *p* < 0.001) in Black SS participants compared to White SS counterparts. Besides, we noted an inverse correlation between plasma IL-18 levels and HDL-C (r = −0.27, *p* < 0.001) and positive correlation with LDL-C (r = −0.33, *p* < 0.001) and plasma triglycerides (r = −0.59, *p* < 0.001) in Black SS participants compared to White SS counterparts. Interestingly, plasma IL-18 levels were inversely correlated to plasma IL-6 levels in Black SS participants but not in White SS counterparts (r = −0.29, *p* < 0.01). 25OHD-S revealed their significant anti-inflammatory effect by inhibition the increase IL-18 plasma level in both SS groups (Figure 3B). Plasma IL-18 level was reduced at D2-36 weeks in the Whites SS group before and after 25OHD-S. At opposite, in the SS Black group, a reduction was observed at D60-36 weeks. Plasma basal Interleukin-23 (IL-23) levels were significantly increased in the Black SS group compared to the White SS group. In our study, we found an inverse correlation between plasma IL-23 levels both with vitamin D deficiency and eGFR drop in Black SS participants, but not in White SS counterparts (r = −0.64, *p* < 0.001; r = −0.47, *p* < 0.001, respectively). Significant positive 25OHD-S effects were observed by decrease IL-23 in both SS groups (Figure 3C). Plasma IL-23 level was declined at D2-36 weeks in the White SS group before and after 25OHD-S. Conversely, in the Black SS group, a reduction was observed at D60-36 weeks.

### 3.3. Effects of 25OHD-S on Oxidative Stress Imbalance Status

#### 3.3.1. Total Antioxidant Status (TAS), Malondialdehyde (MDA) Levels and Antioxidant Enzyme Activities Profile

All Black and White SS groups (Figure 4B) showed a depletion of plasma basal TAS. However, the lipid peroxidation estimates in terms by plasma basal TBARS of MDA levels are extremely high and inversely proportional to the TAS levels in Blacks SS group compared to Whites SS group (Figure 4A). The basal total and erythrocytes SOD activities were decreased proportionally both with an increase of MDA and the TAS depletion (Figure 4C and Figure 4D, respectively). Concomitantly, basal erythrocytes GPx activity (Figure 4E) was significantly reduced in both group, but more marked in the Black SS group. Figure 4F illustrated that Black SS participants exhibited drastically lower depletion of plasma glutathione (GSH) levels compared to Whites SS group. 25OHD-S exerted a significant inhibitory effect on oxidative stress (MDA reduction, TAS increase, SOD and GPx stimulation, GSH enhance), particularly at D2-36 weeks in the Whites SS group compared to Black SS group and at D60-36 weeks of 25OHD-S in Black SS group (Figure 4C, Figure 4D, Figure 4E and Figure 4F, respectively; *p* < 0.0001).

#### 3.3.2. Antioxidant Trace Elements Profile

The basal antioxidant trace elements profile is significantly different between Black and White SS participants. Plasma selenium (Figure 5A) and manganese (Figure 5B) levels are depleted in the Black SS group compared to the White SS group. However, plasma copper (Figure 5C) and zinc (Figure 5D) levels are enhanced in the Black SS group compared to the White SS group. We noted an inverse correlation between plasma selenium levels and TNF-α (r = −0.77, *p* < 0.001) and positive correlation with GPx activity (r = −0.63, *p* < 0.001) and plasma GSH (r = −0.59, *p* < 0.001) in Black SS participants compared to their White SS counterparts. 25OHD-S significantly increased serum selenium levels by 33% (Figure 5A; *p* < 0.0001), manganese by 57% (Figure 5B; *p* < 0.0001), copper by 25% (Figure 5C; *p* < 0.0001) and zinc by 26% (Figure 5D; *p* < 0.0001) compared to baseline levels in both groups at D2-36 weeks in the White SS group and at D60-36 weeks in the Black SS group.

### 3.4. Effects of 25OHD-S on MetS Components

The metabolic syndrome (MetS) clusters of the participants according to race and ethnicity were summarized in Table 2 and Table 3. The results mentioned in Table 4 and Table 5 show a severe 25-OHD deficiency in the Black SS group compared to the White SS group (6.98 ± 1.91 vs. 19.8 ± 4.33 ng/mL, *p* < 0.001). In contrast, the Black SS showed significantly higher serum 1, 25 (OH) 2D compared to White SS (44.3 ± 3.55 vs. 29.9 ± 7.51 pg/mL, *p* < 0.001). We observed a significant positive correlation between 25-OHD deficiency and eGFR decline in both groups, but more pronounced in the Black SS group compared to the White SS group (r = +0.79, *p* < 0.001; r = +0.46, *p* < 0.01, respectively). The 25OHD-S was clinically well tolerated, and no participants exhibited severe adverse effects. Additionally, no participant in both groups developed hypercalcemia. The data displayed in Table 2 and Table 3 characterize the dose–response effect between 25OHD-S on the metabolic biomarkers and serum vitamin D level in Black versus White SS CKD participants. 25OHD-S significantly enhanced both serum 25(OH) D levels and 1, 25(OH) 2D concentrations proportionally to the dose in both White and Black SS groups compared to baseline levels. This rise was immediate and sustained in the White SS group from 24 weeks, but slowly and until 36 weeks in the Black SS group. The result was obtained at 2000 IU (D2) and 60,000 IU (D60), concomitantly at 24 and 36 weeks with both two vitamin D_3_ doses. Surprisingly, we noted that serum 25OHD levels were not normalized in the Black SS group at D2 dose both at 24 and 36 weeks compared to the White SS group. The Black SS groups persist in vitamin D_3_ insufficiency state (<30 ng/mL). Nevertheless, Black SS groups achieved the treatment target before and after vitamin D treatment (>30 ng/mL: 6.98 ± 1.91 vs. 44.9 ± 2.45 ng/mL, *p* < 0.001, respectively) with D60 at 36 weeks where serum 25(OH) D levels and 1, 25(OH) 2D levels are increased by 84% and 53%, respectively (*p* < 0.001). The body mass index (BMI) and the percent body fat in subcutaneous-abdominal adipose tissue were normal in White SS participants while the Black SS are obese (Table 2 and Table 3). The waist circumference (WC) highlights visceral adiposity in the Black SS group compared to the White SS group (Table 2 and Table 3). The BMI was positively correlated with the percentage of body fat in the Black SS group but not in the White SS group (r = +0.49, *p* < 0.01).

The 25OHD-S caused moderately decrease in BMI, WC, WC/WH ratio percent body fat mass in both groups, but this effect is different between White and Black SS groups according cholecalciferol dose and during treatment. We observed a little significant change in the White SS group at D2-36 weeks. Additionally, an anthropometric status was not significantly improvement in the Black SS group at D60-36 weeks. The data presented in Table 2 and Table 3 shows that both groups were nondiabetic with glucose intolerance, but not hyperglycemia (<6.99 mmol/L or <1.26 g/L). However, serum fasting glucose is moderately lower in the Black SS group than in the White SS group. Additionally, glucose tolerance impaired is associated with hyperinsulinism state in both groups, significantly more pronounced in Black SS versus Whites SS participants. Concurrently, the Homa index (Homa-IR), which reflects insulin resistance, is increased in both groups, but significantly higher in Black SS than in Whites SS, which confirms that Black SS have lower insulin sensitivity. We found a significant positive correlation between Homa-IR and WC in the Black SS group but not in the White SS group (r = +0.65, *p* < 0.001). The 25OHD-S data exhibit in Table 2 and Table 3 revealed a significant reduction in fasting glucose, insulinemia and in Homa-IR at D2-36 weeks in White (*p* < 0.05) and at D60-36 weeks in Black SS groups (*p* < 0.01). The data obtained in Table 2 and Table 3 shows that dyslipidemia is only present in the White SS group participants, but not in the Black SS group. Serum TC and TG levels were significantly lower in the Black SS group than in the White SS group. It should be noted that Black SS participants were not treated with statins. Likewise, serum LDL-C levels were also significantly lower in the Black SS group compared to the White SS group. Conversely, serum HDL-C is significantly higher in the Black SS group than in the White SS group. It should be noted that serum HDL-C levels were superior in women than in men in both groups. The results registered in Table 2 and Table 3 show that 25OHD-S appeared to have a beneficial effect on reducing serum total cholesterol, LDL-C, TG levels in both groups. The diminution is significant at D2-36 weeks in the White SS group and significantly reduced at D60-36 weeks in the Black SS group (*p* < 0.02). 

Oppositely, 25OHD-S increases HDL-C levels in both groups. This augmentation is significant at D2-36 weeks in the White SS group (*p* < 0.01) and significantly reduced at D60-36 weeks in the Black SS group (*p* < 0.01). Despite a favorable lipid profile including lower triglycerides, decrease LDL-C and higher HDL-C, the Black SS group showed higher systolic blood pressure and diastolic blood pressure compared to the White SS group. The data mentioned in Table 2 and Table 3 reveal a slight decrease in systolic blood pressure was noted in the White SS group after 25OHD-S at D2-36 weeks. Oppositely, the arterial pressure disorder is not normalized in the Black SS group, and hypertension persisted (>140 mm Hg) after 25OHD-S both at D2–D60 and during the two treatment times. Serum ceruloplasmin depletion is observed in both groups, but is more significant in White SS participants compared to Black SS participants, averaging 38%. 25OHD-S increased ceruloplasmin levels with a positive correlation between serum ceruloplasmin concentrations and plasma copper levels (r = + 0.75, *p* < 0.001) at D2-36 weeks in the White SS group and at D60-36 weeks in the Black SS group.

### 3.5. Effects of 25OHD-S on CKD Clusters

In this study, the Black and White SS participants were advanced CKD, but not in End Stage Renal Disease, and not requiring dialysis. The data presented in Table 4 and Table 5 indicate that both groups were in CKD stage 3A, defined by eGFR between 45–59 mL/minute per 1.73 m^2^. Compared to White SS, Black SS had significantly lower eGFR. The 25OHD-S had a significant advantageous effect on rising eGFR and slows down its decline in both groups. This augmentation is significant at D2-36 weeks (+21%, *p* < 0.01) in the White SS group and significantly increase at D60-36 weeks (+26%, *p* < 0.01) in the Black SS group. The data mentioned in Table 4 and Table 5 show that serum creatinine and uric acid were significantly increased in Black SS compared to White SS. However, serum albumin and calcium levels were moderately reduced in Black SS compared to White SS, but not significantly. 

The 25OHD-S data displayed in Table 4 and Table 5 illustrate a significant decrease in serum creatinine, creatinine-BMI ratio and uric acid in both groups. The reduction is significant at D2-36 weeks in the White SS group (*p* < 0.01) and significantly reduced at D60-36 weeks in the Black SS group (*p* < 0.01). Inversely, the 25OHD-S increases serum albumin, calcium and ionized calcium. This elevation is significant at D2-36 weeks in the White SS group (*p* < 0.02) and significantly improved at D60-36 weeks in the Black SS group (*p* < 0.01). Compared to White SS (Table 4 and Table 5), Black SS had significantly higher urinary albumin, creatinine, ionized calcium levels and uric acid. The levels of 24 h urine creatinine, calcium and uric acid increased significantly after 25OHD-S in both SS groups (Table 4 and Table 5). The increased excretion of these urinary metabolites is more marked at D2-36 weeks in the Whites SS group; while it is more significant at D60-36 weeks in the Black SS. At opposite, the 25OHD-S decreased significantly urinary albumin at D2-36 weeks in the Whites SS group (−35%, *p* < 0.001); and significantly dropped at D60-36 weeks in the Black SS group (−48%, *p* < 0.001). Serum phosphorus levels were comparable between the two SS groups (Table 2 and Table 3). However, urinary phosphorus levels were reduced in the Black SS group compared to the White SS group (221 ± 17 vs. 302 ± 25 mmol/24 h, *p* < 0.01, respectively). The difference is not significant before and after cholecalciferol supplementation in both groups.

### 3.6. Effects of 25OHD-S on Atherothromboembolic Risk

The data obtained on the atherothromboembolic risk (Table 4 and Table 5) show moderate hyperhomocysteinemia in the White SS group, but slightly higher in the Black SS group. Serum lipoprotein (a) is abnormally elevated only in the Black SS group compared to the White SS group. In this study, hyperhomocysteinemia is not correlated with systolic or diastolic blood pressure in both groups. However, hyperhomocysteinemia is positively correlated with increase Hs-CRP and TNF-alpha in the Black SS group compared to the White SS group (r = +0.72, *p* < 0.001; r = +0.61, *p* < 0.001, respectively). We observed significant positive correlation between higher Lp (a) levels and strongly reduced eGFR in the Black SS group (r = +0.58, *p* < 0.01), but not in the White SS group. Regarding ApoA_1_ and ApoB_100_, we did not find any significant difference between Black and White groups; however, it appears that ApoB_100_ levels were slightly higher in the Black SS group than in the White SS group. Interestingly, we noted significant positive correlation between elevated Lp(a) and increase Apo B_100_–Apo A_1_ ratio compared to the White SS group. The 25OHD-S attenuated atherothrombogenic risk by reduction serum lipoproteinemia (a) and homocysteine levels. However, hyperhomocysteinemia (>12 µmol/L) is maintained in both SS groups after vitamin 25OHD-S (Table 4 and Table 5). Contrariwise, the hyper Lp (a) is normalized in the White SS group compared to the Black SS group (<75 nmol/L). The data shows a slow decrease at D2-36 weeks in the White SS group (*p* < 0.05) and moderate decrease at D60-36 weeks in the Black SS group (*p* < 0.05). Interestingly, 25OHD-S reduces significantly Apo B_100_–Apo A_1_ ratio in the Black SS group at D60-36 weeks (−31%, *p* < 0.001) and moderately in the White SS group at D2-36 weeks (−17%, *p* < 0.001). This effect is exerted by lowering Apo B_100_ (−21%, −9%, respectively, in Black and White SS groups) and increasing Apo A_1_ (+13%, +10%, respectively, in Black and White SS groups), but the difference is not significant.

## 4. Discussion

Based on clinical data, it is confirmed that cholecalciferol supplementation (25OHD-S) modulates the cytokine storm, oxidative stress damage and metabolic disorders. According to the racial–ethnic factor, it appears that 25OHD-S provides a significant benefit both in Black and White SS participants by stabilizing the renal disease and preventing its progression to end-stage renal disease. However, this investigation reveals that intermittent high dose (60,000 IU/month) during long time (36 weeks) in Black SS is efficient compared to White SS subjects with lower effective dose (2000 IU/day) during short time (24 weeks).

The first important point is linked to the dose-duration of 25OHD-S and CKD progression. The results were strongly correlated with the variation in total serum vitamin D (25-OHD and 1,25OH2D) considering the racial and ethnic disparities. Our findings showed drastic depletion of serum 25OHD levels in Black SS participants compared to White SS participants. The probable explanation of this result can be argued by more recent investigations carried out in the Black population such as African Americans on Kidney Disease related to vitamin D deficiency [64] and Hypertension [65]. Previous studies have highlighted 25OHD deficiency in Black people is associated with increased melanin should be decreased decrease dermal synthesis of vitamin D from 7-dehydrocholesterol due to increase the light energy photons absorption, but their clinical implications remain unclear [66]. In Black East Africa people, the melanin pigment is an natural absorbent of the UV-B radiation, and therefore dark-skinned people require substantially more exposure to UV-B rays than fair skinned people in order to synthesize an adequate amount of 25OH D (vitamin D_3_). Thus, dark-skinned people, such as Algerian Black SS, are more susceptible to developing 25OH D deficiency in the absence of adequate sunlight exposure [67]. Besides, this difference could result from feedback inhibition of hepatic synthesis of 25-OHD by the increased circulating 1,25(OH) 2D in the Black SS people [68]. Our randomized and controlled SS study was shown an interaction between race and dose–response not similarly to that seen between White and Black SS participants. Indeed, the optimal cholecalciferol supplementation dose required to normalize serum 25OHD levels was obtained at 2000 IU/24 weeks in White SS group. Conversely, the optimal 25OHD-S dose required to achieve target serum 25OHD level ≥75 nmol/L was obtained at 60,000 IU/36 weeks in the Black SS group, but not at 2000 IU/24 weeks. Several studies have shown that Black subjects require therapeutic doses of vitamin D higher than 800 IU/day to obtain physiological effects compared to White subjects, who require lower doses [69].

The second crucial point concerned the CKD-inflammation induced by pro-inflammatory cytokines (PIC) storm and immunomodulating state exerted by 25OHD-S. Some studies have demonstrated in CKD patients a direct relationship between uremic toxins and PIC overproduction from neutrophils and monocytes may induce the TLR4 (Toll-like receptor-4) activation [70]. The TLR4 expression may activate the intracellular NF-κB (nuclear factor-κB) pathway and enhances the expression of NF-κB-controlled genes of PIC [71]. Among PIC, TNF-α was found to be very high in Black SS participants compared to White SS participants (Figure 2C). TNF-α is recognized to play a central role in renal inflammation and fibrosis via acting as a stimulant for the release of MCP-1 or CCL2 (monocyte chemoattractant protein-1 or chemokine ligand 2). Previous study describe that TNF-α is expressed principally by macrophages in renal mesangial cells, promoting inflammation across NF-Kb signaling pathways [72]. Interestingly, vitamin D inhibits the synthesis of NF-kB pathway [73], which may explain the decline significantly plasma TNF*-α* level after 25OHD-S. Concerning other PIC, in our study, we have observed that IL-1β, IL-6, IL-18, IL-12 and IL-23 evolve similarly to TNF*-α* in Black SS participants versus White SS participants (Figure 2A,B and Figure 3A–C). IL-18 is the cytokine more specific to progression of the CKD, where its expression is positively correlated to albuminuria [74]. IL-6 is involved in the differentiation of Th17 lymphocytes in the presence of the growth factor TGFβ (Transforming growth factor beta). IL-12 regulates the differentiation of native T cells into Th1 cells, which is determining a protective early immune response to renal inflammation, such as peritonitis [75]. Recently, it was reported that IL-23 is involved in kidney differentiation of Th17 cells in a pro-inflammatory and mainly in the presence of TGF-β [76], has been proposed as a key mediator of kidney tissue damage concomitantly with IL-17, and opens new therapeutic targets in inflammatory kidney disease [77]. It has been described that IL-1β increased expression of NGAL (neutrophil gelatinase-associated lipocalin), a biomarker for kidney injury [78]. Likewise, the receptor of IL-1β is associated with IRAK-M (Interleukin 1 Receptor associated Kinase 3Macrophages), a biomarker for renal atrophy and interstitial fibrosis. IRAK-M served as an inhibitory signal for IL-1β-mediated signaling in macrophages, indicating that IL-1β activated macrophages toward a pro-inflammatory phenotype augmenting kidney tissue damage [79]. Regarding the immunomodulation effects exerted by 25OHD-S on PIC, the conclusions are often controversial, either with an inhibitory effect on cytokines production [80] or contrary, with stimulating effects [81]. In our investigation, we showed that 25OHD-S attenuated the inflammatory state by significantly reduced the serum CRP-us levels and PIC overproduction in both Black and White SS groups. Our data can be explained by a lymphocytes proliferation effect. Indeed, Vitamin D stimulates the maturation of Th17 which are considered to protect the cellular integrity barrier [82]. Accurately, 25OHD-S actions are argued across the VDR (vitamin D receptor) present on immune cells, such as monocytes and T lymphocytes. The VDR expression influences significantly the differentiation and proliferation of these immune cells [83]. Besides, the vitamin D supplementation was able to reduce the expression of TRL4, cathelicidin (endogenous antimicrobial peptide), and MCP-1 in the uremic environment [84]. On the other hand, it is important to add that uremia suppresses immune signal-induced CYP27B1 (cytochrome P450 family 27 subfamily B member 1) encoding for 1 α -hydroxylase, which is expressed primarily in kidney proximal tubular epithelial cells, but also in extra-renal tissues, such as immune cells. This enzyme converts vitamin D into its active form, 1,25-dihydroxyvitamin D_3_ or calcitriol. In the T cells, calcitriol influences T cell function in intracrine and paracrine endocrine effects by conversion of 25(OH)D to 1,25(OH)2D3 [85].

The third major point concerned the relationship between oxidative stress damage due to the drop of anti-oxidant system, such as SOD, GPx, selenium (Se), manganese (Figure 5B), copper (Figure 5C), zinc (Figure 5D) and vitamin D deficiency in both Black and White SS groups. These events enhance the ROS production and increased circulating levels of malondialdehyde, which depletes the TAS (Figure 4A–D). Our data are in agreement with many recent studies in patients with CKD [14,15,16,19]. It should be emphasized that a deleterious cycle is established between increasing oxidative stress leading to increased inflammation in the CKD, which can lead to atherosclerosis [86]. In this investigation, our attention was focused particularly on Se. There have been a few studies focusing on Se in CKD. In this study, Se represents the link between hCys, oxidative stress (GPx activity), and inflammation (TNF-α, IL-6) during CKD evolution. In our study, we showed that Se plasma levels (Figure 5A) and erythrocytes GPx activities are altered in Black SS CKD participants compared to Whites SS group (Figure 4E,F). We have observed a gradual decrease plasma GPx activity and high serum creatinine in Black SS group compared to White SS participants. Although GPx is synthesized by different tissues, renal proximal tubular epithelial cells are the main source of this enzyme, therefore presenting a challenge to the renal function that could result in lower enzyme production [87]. The kidneys play a central role in the homeostasis of Se. In fact, the kidneys and thyroid gland have the highest Se concentration of all human organs [88]. On the other hand, GPx and Se association may be used to counteract the effect of increased peroxides, which accompanies renal damage. Moreover, the presence of toxin complexes in the blood of CKD patients may inhibit the activity of GPx [89]. Moreover, previous studies have demonstrated that decreased GPx activity in erythrocytes leads to accumulation of hydrogen peroxide related to lipid peroxidation that may cause inhibition of SOD activity [90]. Interestingly, Se exerts immunomodulatory and anti-inflammatory properties. Indeed, we showed that Se deficiency is correlated to higher levels of inflammatory cytokines, particularly with TNF-α and IL-6 significantly in Black SS CKD participants compared to the White SS group. This observation seems to be explained by the suppression of selenoprotein expression [91]. Regarding vitamin D supplementation actions, in our study, 25OHD-S attenuates the deleterious effects of oxidative stress by activating enzymatic (SOD, GPx) and trace element antioxidant protection. A recent study showed that vitamin D acts indirectly on oxidative stress signaling pathways dependent on inflammation pathways, mainly via the NF-κB pathway [92]. 25OHD-S increases Se concentrations via enhancement selenoproteins synthesis and attenuates pro-inflammatory gene expression in macrophages [93]. In this context, the relationship between the effects of 25OHD-S and zinc has attracted our attention, since this trace elements is involved both in the maturation of Th1 lymphocytes [94], and in the oxidative stress (SOD activity). Interestingly, the copper/zinc ratio is modified by vitamin D supplementation in Black SS subjects compared to White SS subjects. Some works have described that the elevation of the copper/zinc ratio (case of our study) is associated with the aggravation of oxidative stress, inflammation, malnutrition and immune deficiency. In the blood circulation, 95% of copper is transported by ceruloplasmin [95]. In our study, we observed a drastic drop in serum ceruloplasmin concentrations in the White SS group compared to the Black SS group. This argument may partly explain the accumulation of serum copper and its reduced availability for SOD activity in the White SS group compared to the Black SS group. The decrease in serum copper concentrations under treatment with vitamin D can be argued by the enhancement of its transport protein, ceruloplasmin, which will result in a better bioavailability of Copper, having a favorable impact on total and erythrocyte SOD-Cu/Zn activity [96].

Regarding PIC interactions with oxidative stress and atherothroboembolic risk (hCys, Lp (a)), we found that IL-6, TNF-α, and hs-CRP were significantly correlated with elevation in total plasma hCys both in Black and White SS participants. Previous studies have shown that TNF-α and IL-6 have atherogenic properties concomitantly with increased hCys and lead to endothelial dysfunction by vascular calcification [97]. In this context, vitamin D deficiency is linked with increased hCys in CKD through vascular endothelial cells express VDR and 1α-hydroxylase synthesis [98]. As regards vitamin D supplementation, in our study, 25OHD-S did not attenuate hyper hCys in both Black SS and White SS groups. This is possible due to the character of hCys as an independent factor that cannot be influenced by vitamin D. As previously argued, we observed the highest plasma hCys levels concomitantly with drastic depletion both plasma Se levels and GPx activity in the Black SS group compared to the White group. Previous studies have shown that Black American compared to White American people shown Se deficiency and a defect in the synthesis of Se transport proteins, such as selenoproteins [99]. Indeed, Se exerts its antioxidant effects being integrated into selenoproteins, such as Sepp1 (selenoprotein P1), and SelS (selenoprotein S). Sepp1 is a major selenoprotein in tissues serving as a reservoir of Se. The lack of synthesis of Sepp1 can lead to hyper hCys, which in turn inhibits the synthesis of glutathione (decline of GSH in our study), and therefore decreases GPx activity. Recently, it has been proven that these two events (simultaneously deficiency in Se and GPx) lead to selenoproteins depletion and to reduce concomitantly the renal antioxidant and anti-inflammatory protection [100].

Concerning the abnormal Lp (a) levels, this fact can be explained by the drop of eGFR, which increases Lp (a) levels in the Black SS group compared to the White SS group, because it is eliminated essentially by glomerular filtration [101]. In this context, an association has been described between increase Lp (a) levels and hypertension through renal endothelial dysfunction and oxidative stress. Indeed, an oxidative process that makes Lp (a) oxidized leads to the development of atheromatous plaques [102], and generates other oxidized metabolites, which will interact with the Toll-like receptors, particularly in the Black population [103]. In addition, the resurgence of Lp (a) in CKD subjects may also be linked to drug treatment where Lp (a) is very little reduced by statins [104]. As previously argued for hCys, vitamin D supplementation did not attenuate high Lp (a) levels in both SS groups. This observation explains the atherothromboembolic event in the Black SS group. Little research has studied the vitamin D effects on Lp (a). However, it was suggested that 25OHD-S can be decreased serum Lp (a) via increase apoprotein A1 [105]. This event can explain the decreasing the Apo B_100_–Apo A_1_ ratio in the White SS group, but not in the Black SS group, under 25OHD-S. In this context, numerous experimental studies have shown a relationship between the activation of VDR and increased expression of ApoA1, through modulation of SREBP2 (sterol responsive element binding protein 2) [106].

It is important to argue the effects of vitamin D supplementation on anthropometric parameters, particularly on fat mass. We have shown that 25OHD-S is positively correlated with significantly BMI and BF% attenuation in the White SS group and to reduce abdominal adiposity, but moderately in the Black SS group [107]. However, the significant changes in BMI and BF% cannot be attributed solely to vitamin D supplementation. Indeed, it seems that in Black SS subjects, the association of vitamin D–Calcium supplementation is more effective than vitamin D alone. As mentioned in Table 3 and Table 4, the results can be explained by increases of the ionized calcium (iCa) in the Black SS subjects compared to the White SS counterparts. Thus, the sequestration of vitamin D in the adipose tissue of Black subjects in the presence of calcium can activate lipolysis signaling pathways. Overall, there is consistent evidence that calcium and vitamin D increase whole body fat oxidation and the lipolytic effects of calcium are admitted in some studies [108]. As several studies and other trials show, that vitamin D supplementation may be significantly associated with less weight gain, but this association may be dependent on adjunctive calcium supplementation and a particular of fat region [109,110].

Our results are in agreement with several studies carried in the Black African American subject linked to a lower resting metabolic rate [111]. This study highlight elevated obesity prevalence in the black population [112]. BF% reduction under vitamin D supplementation can also be explained via the effects of leptin, which promotes the drop of caloric intake and increases basal energy expenditure. We are in agreement with several studies on this subject carried in the Black African American subjects. However, the data are controversial about vitamin D’s effect and decrease of BMI. The Black subject requires high doses of vitamin D to have physiological benefits compared to the White subject, who requires low doses of vitamin D [113].

In our investigation, it is probable that 25OHD deficiency in obese Black SS participants is due to sequestration of vitamin D in adipose tissue (25OHD is lipophilic soluble vitamin), leading to a decrease in bioavailability and a decrease in circulating 25OHD [114]. In addition, the results obtained from the food questionnaire showed that White subjects show a low food intake with a better nutritional density compared to Black subjects. In the nutritional context, it has been shown that a low-energy diet applied in patients with nephropathy may decrease serum resistin levels and inflammation. In addition, responses of all adipokines to dieting appear to be affected by body fat mass, especially android fat mass [115]. A previous study has shown that vitamin D administration is associated with an increase in adiponectin and a decrease in leptin level in end-stage renal disease patients [116].

Regarding phosphorus and calcium profile, Blacks SS participants appear to have more efficient calcium compared to White SS participants despite a very lower serum 25OHD level in Blacks SS group. In agreement with our findings, results from a recent study showed that efficiency of calcium absorption correlated significantly with serum 25OHD, but also with intestinal transit, and urinary calcium-to-creatinine ratio. In addition, the reduction in urinary calcium in the Black SS participants results from diminished intestinal absorption of calcium linked to the increases in serum parathyroid hormone (PTH), serum 1,25(OH)2D, and possibly enhances in urinary cAMP (cyclic adenosine monophosphate). Paradoxically, increased circulating PTH in Black people prevents urinary loss of calcium by enhancing the tubular reabsorption [117]. It is likely that intestinal absorption of phosphate may be diminished in Black SS versus White SS participants. It could be explained that increases both in plasma PTH and 1,25(OH)2D, and the decrease in urinary calcium results from reduced intestinal absorption of calcium and phosphorus. This observation was reported previously and may result from low plasma renin and aldosterone in Black subjects [118]. In our study, cholecalciferol supplementation do not increases the absorption of phosphorus, which did not lead to deleterious hyperphosphatemia [119].

## 5. Conclusions

This study highlights that the severity of vitamin D deficiency is dependent on racial–ethnic factors; it is more acute in Black SS subjects than in White SS subjects. Vitamin D_3_ (Cholecalciferol) supplementation in CKD subjects is beneficial both in Black SS and White SS counterparts by stabilizing the renal disease by inhibiting the deleterious effects of both pro-inflammatory cytokines storm and oxidative stress damage to prevent its progression to end-stage renal failure and renal dialysis. Cholecalciferol supplementation may contribute to the improvement of CKD evolution, but does not substitute medical therapy.

## Figures and Tables

**Figure 1 nutrients-14-02285-f001:**
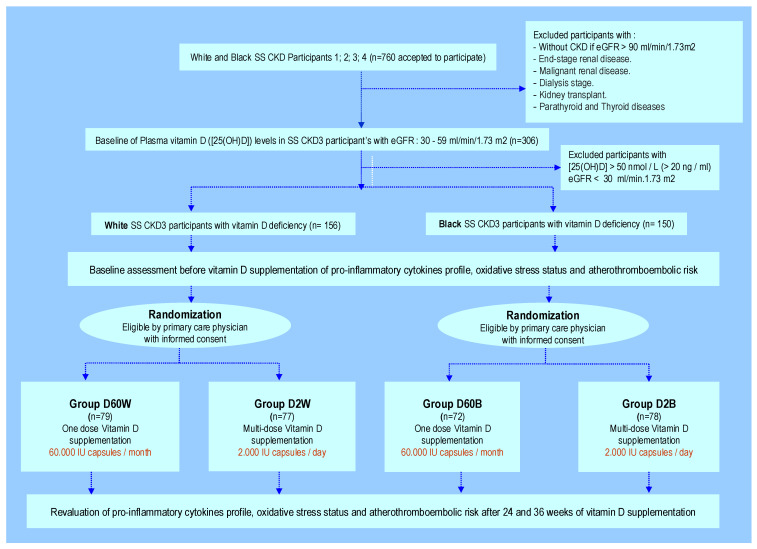
Algorithm clinical protocol randomization of vitamin D_3_ supplementation in CKD Southern Sahara (SS) White (W) and Black (B) people at 2000 IU doses every day (D2 group) versus 60,000 IU doses every month (D60 group) during 24 and 36 weeks vitamin D_3_ treatment. eGFR: estimated Glomerular filtration rate; The numbers 1, 2, 3 (A and B) and 4 represent the four stages of CKD on the basis of eGFR: Stage 1 with normal or low eGFR (GFR < 90 mL/min per 1.73 m^2^), Stage 2 with mild CKD (GFR = 60–89 mL/min per 1.73 m^2^), Stage 3A with moderate CKD (GFR = 45–59 mL/min per 1.73 m^2^), Stage 3B with severe CKD (GFR = 30–44 mL/min per 1.73 m^2^) and Stage 4 with morbid CKD (GFR = 15–29 mL/min per 1.73 m^2^).

**Figure 2 nutrients-14-02285-f002:**
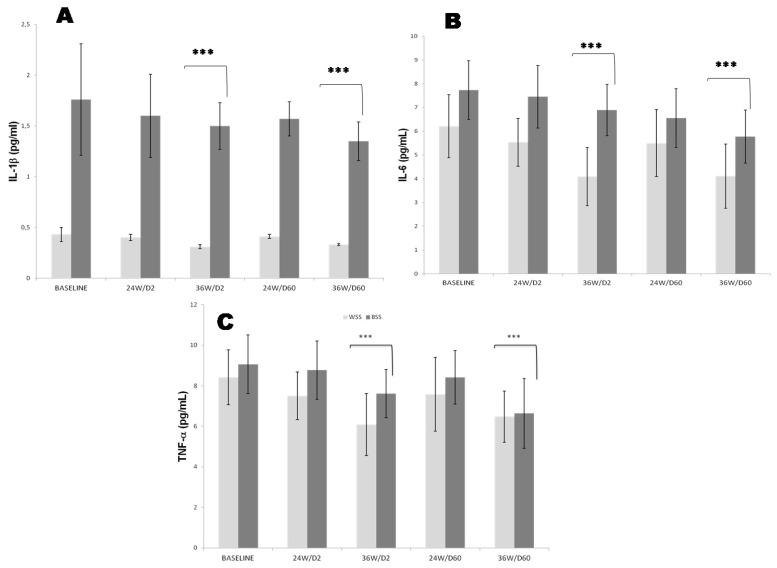
Effects of oral 2000 IU/day and 60,000 IU/month vitamin D_3_ supplementation on plasma **C:** TNF-α (tumor necrosis factor-α*),* (**A**): IL-1β (interleukin-1 beta) and (**B**): IL-6 (interleukin-6) in CKD White and Black SS people at 2000 IU doses every day (D2 group) versus 60,000 IU doses every month (D60 group) during 24 and 36 weeks vitamin D_3_ treatment. The mean values are assigned from the standard error to the mean (X ± ESM). The degree of significance is calculated for a risk of error α = 5%. The mean comparison is established for each group, White SS and Black SS. Baseline data were obtained before vitamin D supplementation. *p*-value is calculated at baseline time (White SS versus Black SS groups). *** *p* < 0.001.

**Figure 3 nutrients-14-02285-f003:**
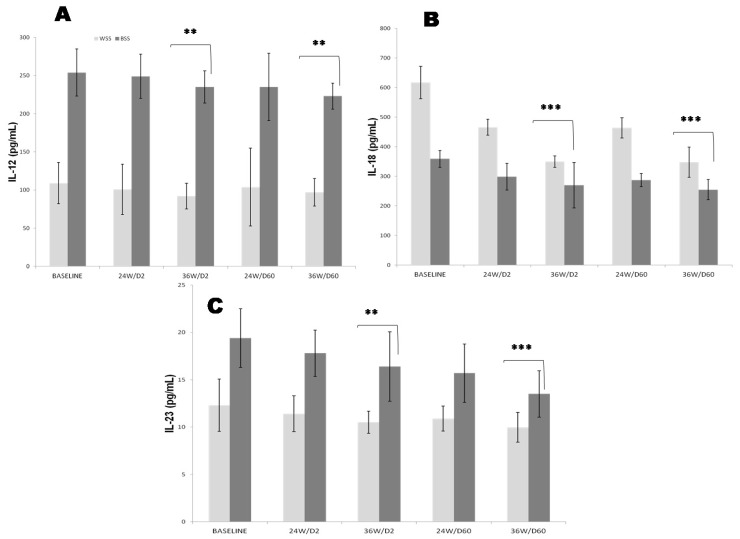
Effects of oral 2000 IU/day and 60,000 IU/month vitamin D_3_ supplementation on plasma (**A**): IL-12 (interleukin-12), (**B**): IL-18 (interleukin-18) and (**C**): IL-23 (interleukin-23) in CKD White and Black SS people at 2000 IU doses every day (D2 group) versus 60,000 IU doses every month (D60 group). The mean values are assigned from the standard error to the mean (X ± ESM). The degree of significance is calculated for a risk of error α = 5%. The mean comparison is established for each group, White SS and Black SS. Baseline data were obtained before vitamin D supplementation. *p*-value is calculated at baseline time (White SS versus Black SS groups). ** *p* < 0.01; *** *p* < 0.001.

**Figure 4 nutrients-14-02285-f004:**
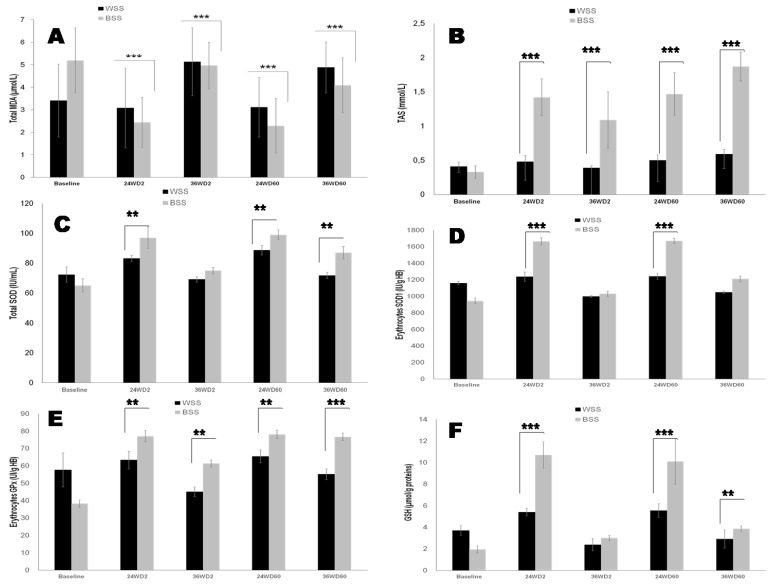
Effects of oral 2000 IU/day and 60,000 IU/month vitamin D_3_ supplementation on serum oxidative stress biomarkers in Black versus White SS CKD participants. (**A**): Blood total MDA (Malondialdehyde); (**B**): Blood TAS (Total Antioxidant Status); (**C**): Total SOD (total Superoxide dismutase activity); (**D**): Erythrocytes SOD 1; (**E**): Erythrocytes GPx (Glutathione peroxidase); (**F**): serum GSH (glutathione). The mean values are assigned from the standard error to the mean (X ± ESM). The degree of significance is calculated for a risk of error α = 5%. The mean comparison is established for each group, White SS and Black SS. Baseline data were obtained before vitamin D supplementation. *p*-value is calculated at baseline time (White SS versus Black SS groups). ** *p* < 0.01; *** *p* < 0.001.

**Figure 5 nutrients-14-02285-f005:**
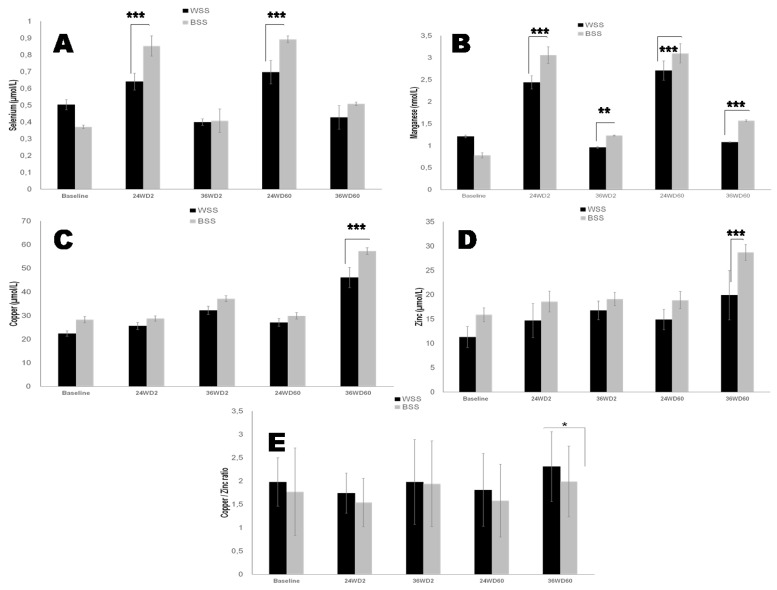
Effects of oral 2000 IU/day and 60,000 IU/month vitamin D_3_ supplementation on serum trace elements profile in Black versus White SS CKD participants. (**A**): Selenium; (**B**): Manganese; (**C**): Copper; (**D**): Zinc; (**E**): Copper–Zinc ratio. The mean values are assigned from the standard error to the mean (X ± ESM). The degree of significance is calculated for a risk of error α = 5%. The mean comparison is established for each group, White SS and Black SS. Baseline data were obtained before vitamin D supplementation. *p*-value is calculated at baseline time (White SS versus Black SS groups). * *p* < 0.05; ** *p* < 0.01; *** *p* < 0.001.

**Table 1 nutrients-14-02285-t001:** Baseline Clinical characteristics in Blacks versus Whites in SS general population.

Parameters	Whites (*N* = 79)	Blacks (*N* = 72)	*p* Value-Baseline
Skin colour (%)	46.3	53.7	<0.01
Age (year)	52 ± 3	49 ± 1	0.072
Gender (%)	48/33 ^(F/M)^	53/41 ^(F/M)^	0.083
CKD3 (%)	37	49	<0.001
Comorbidities (%)			
Hypertension	60.5	74.9	<0.001
Anemia	66.6	57.2	<0.01
Coronary artery disease	45	32	<0.01
Congestive heart failure	19	25	<0.01
Cerebral vascular accident	15	9	<0.01
Peripheral vascular disease	19	12	<0.01
Obesity	60	86	<0.01
Dyslipidemia	75	28	<0.01
Hypocalcaemia	42.8	60.6	<0.01
Hyperuricemia	22.5	36.5	<0.01
Smoking current (%)	20	18	0.091
Alcohol use (%)	2	1	0.073
Current drug use (%)			
Calcium channel blocker	45	52	0.144
β-Blockers	45	46	0.061
Thiazide	7	10	0.205
Diuretic	63	76	0.331
Aspirin	56	58	0.069
Statins	28	39	0.177
Vitamin D intake (µg/day)	69 ± 4	97 ± 3	<0.001
Vitamin D intake (IU/day)	2760 ± 160	3880 ± 120	<0.001
Calcium intake (mg/day)	977 ± 11	691 ± 22	<0.01
Caloric intake (kcal/kg/day)	35 ± 5	62 ± 9	<0.001
Physical activity (min/week)	60 ± 5	40 ± 3	<0.01

F: female; M: male; CKD: Chronic Kidney Disease stage; SS: Southern Sahara; *p*: significance degree. Data are reported as mean ± S.D (Standard deviation) or proportions. Data of vitamin D and calcium intake are given before oral vitamin D supplementation. One microgram of vitamin D is equal to 40 IU.

**Table 2 nutrients-14-02285-t002:** Effects of oral 2000 IU vitamin D3 supplementation/day at 24 and 36 weeks on the MetS clusters in Blacks versus Whites SS CKD participants.

	Baseline	24 Weeks	24 Weeks		36 Weeks		*p* Value
Parameters	WSS (*N* = 79)	BSS (*N* = 72)	WSS (*N* = 74)	BSS (*N* = 70)	WSS (*N* = 71)	BSS (*N* = 68)	
Body Weight (Kg)	81.2 ± 12.3	93.3 ± 3.61	78.6 ± 6.42	91.9 ± 7.60	76.9 ± 3.30	87.3 ± 5.80	<0.01
BMI (kg/m^2^)	28.1 ± 4.25	32.3 ± 1.26	27.2 ± 2.22	31.8 ± 2.63	26.6 ± 1.15	30.2 ± 1.99	<0.01
WC (cm)	84.9 ± 3.98	107 ± 6.90	83.9 ± 5.55	100 ± 3.89	82.5 ± 4.11	101 ± 2.11	<0.001
WC/WHratio	1.09 ± 0.02	1.13 ± 0.02	1.05 ± 0.01	1.11 ± 0.04	1.03 ± 0.03	1.09 ± 0.05	<0.01
BF (%)	46.6 ± 2.17	55.1 ± 8.12	45.9 ± 3.32	54.5 ± 4.11	39.3 ± 7.71	53.9 ± 2.41	<0.001
Glycemia (mmol/L)	6.70 ± 1.49	5.91 ± 1.97	6.38 ± 1.09	5.66 ± 1.83	5.67 ± 1.11	5.49 ± 1.33	0.223
Insulinemia (pmol/mL)	81 ± 7.15	114 ± 2.91	77.7 ± 3.55	105 ± 5.62	72.9 ± 5.76	95.6 ± 4.79	<0.001
HOMA-IR	3.50 ± 0.93	5.01 ± 0.67	3.17 ± 0.11	4.50 ± 0.55	2.64 ± 0.34	4.09 ± 0.26	<0.001
Triglycerides (mmol/L)	2.19 ± 0.28	1.45 ± 0.26	1.84 ± 0.71	1.32 ± 0.31	1.65 ± 0.05	1.15 ± 0.01	<0.001
Total Cholesterol (mmol/L)	5.84 ± 0.68	4.75 ± 0.32	5.39 ± 0.88	4.26 ± 0.43	5.12 ± 0.33	3.82 ± 0.81	<0.001
HDL-C (mmol/L)	0.85 ± 0.08 ^(M)^	0.95 ± 0.18 ^(M)^	1.14 ± 0.3 ^(M)^	1.09 ± 0.17 ^(M)^	1.24 ± 0.04 ^(M)^	1.25 ± 0.04 ^(M)^	<0.01
	1.11 ± 0.15 ^(F)^	1.24 ± 0.13 ^(F)^	1.21 ± 0.8 ^(F)^	1.27 ± 0.05 ^(F)^	1.26 ± 0.03 ^(F)^	1.28 ± 0.02 ^(F)^	<0.01
LDL-C (mmol/L)	4.52 ± 0.81	3.17 ± 0.79	4.29 ± 0.27	3.02 ± 0.44	4.06 ± 0.16	2.66 ± 0.23	<0.001
Hs-CRP (mg/L)	5.90 ± 1.11	8.18 ± 1.33	5.07 ± 1.44	8.06 ± 1.01	4.45 ± 1.09	7.81 ± 1.68	<0.01
Fibrinogen (g/L)	4.28 ± 1.15	4.44 ± 1.27	4.31 ± 1.22	4.52 ± 1.13	4.25 ± 1.09	4.49 ± 1.11	0.457
Ceruloplasmin (µmol/L)	1.04 ± 0.32	1.68 ± 0.57	1.90 ± 0.99	1.79 ± 0.33	2.44 ± 0.16	1.91 ± 0.33	<0.001
SBP (mmHg)	141 ± 3	159 ± 7	135 ± 1	157 ± 2	130 ± 5	155 ± 6	<0.01
DBP (mmHg)	72 ± 3	90 ± 2	70 ± 5	87 ± 1	68 ± 3	85 ± 2	0.145
S-Phosphorus (mmol/L)	1.20 ± 0.19	1.19 ± 0.11	1.15 ± 0.81	1.16 ± 0.44	1.12 ± 0.73	1.14 ± 0.55	0.117
U-Phosphorus (mmol/24 h)	302 ± 25	221 ± 17	272 ± 51	211 ± 33	254 ± 13	209 ± 47	<0.01

W: Whites; B: Black; CKD: Chronic Kidney Disease stage; SS: Southern Sahara; Metabolic Syndrome (MetS); F: female; M: male; BMI: Body mass index; WC: Waist Circumference; WH: Waist Hips; BF: percent body fat; HOMA: Homeostasis Model Assessment; C: cholesterol; HDL: high density lipoprotein; LDL: low density lipoprotein; SBP: systolic blood pressure; DBP: diastolic blood pressure; Hs-CRP: High sensitive C reactive Protein. The mean values are assigned from the standard error to the mean (X ± ESM). The degree of significance is calculated for a risk of error α = 5%. The mean comparison is established for each group, White SS and Black SS. Baseline data were obtained before vitamin D supplementation. *p* -value is calculated at baseline time (White SS versus Black SS groups).

**Table 3 nutrients-14-02285-t003:** Effects of oral 60,000 IU vitamin D3 supplementation/month at 24 and 36 weeks on the MetS clusters in CKD Blacks versus Whites SS participant’s.

	Baseline	24 Weeks			36 Weeks		*p* Value
Parameters	WSS (*N* = 79)	BSS (*N* = 72)	WSS (*N* = 74)	BSS (*N* = 70)	WSS (*N* = 71)	BSS (*N* = 68)	
Body Weight (Kg)	81.2 ± 12.3	93.3 ± 3.61	78.6 ± 9.11	89.9 ± 4.16	77.2 ± 7.31	86.1 ± 6.11	<0.01
BMI (kg/m^2^)	28.1 ± 4.25	32.3 ± 1.26	27.2 ± 3.11	31.1 ± 1.37	26.7 ± 2.51	30.1 ± 2.08	<0.01
WC (cm)	84.9 ± 3.98	107 ± 6.90	83.4 ± 3.74	103 ± 2.98	82.5 ± 3.21	99.1 ± 3.77	<0.001
WC/WH ratio	1.09 ± 0.02	1.13 ± 0.02	1.05 ± 0.02	1.11 ± 0.03	1.02 ± 0.01	1.01 ± 0.02	<0.01
BF (%)	46.6 ± 2.17	55.1 ± 8.12	45.7 ± 2.23	53.7 ± 3.21	40.1 ± 5.18	52.1 ± 3.14	<0.001
Glycemia (mmol/L)	6.70 ± 1.49	5.91 ± 1.97	6.55 ± 1.17	5.46 ± 1.37	5.69 ± 1.71	5.38 ± 1.24	0.251
Insulinemia (pmol/mL)	81 ± 7.15	114 ± 2.91	73.5 ± 2.81	92.1 ± 4.26	71.5 ± 3.67	80.9 ± 3.92	<0.001
HOMA-IR	3.50 ± 0.93	5.01 ± 0.67	3.10 ± 0.14	3.80 ± 0.27	2.60 ± 0.43	3.30 ± 0.63	<0.001
Triglycerides (mmol/L)	2.19 ± 0.28	1.45 ± 0.26	1.80 ± 0.77	1.21 ± 0.13	1.68 ± 0.08	1.09 ± 0.02	<0.01
Total Cholesterol (mmol/L)	5.84 ± 0.68	4.75 ± 0.32	4.94 ± 0.77	4.31 ± 0.35	4.40 ± 0.22	3.91 ± 0.17	<0.001
HDL-C (mmol/L)	0.85 ± 0.08 (^M^)	0.95 ± 0.18 (^M^)	1.12 ± 0.01 (^M^)	1.18 ± 0.11 (^M^)	1.24 ± 0.04 (^M^)	1.27 ± 0.05 (^M^)	<0.01
	1.11 ± 0.15 (^F^)	1.24 ± 0.13 (^F^)	1.25 ± 0.05 (^F^)	1.28 ± 0.05 (^F^)	1.26 ± 0.03 (^F^)	1.30 ± 0.04 (^F^)	<0.01
LDL-C (mmol/L)	4.52 ± 0.81	3.17 ± 0.79	4.01 ± 0.73	3.08 ± 0.31	3.61 ± 0.74	2.63 ± 0.33	<0.001
Hs-CRP (mg/L)	5.90 ± 1.11	8.18 ± 1.33	5.01 ± 1.54	6.45 ± 1.23	4.24 ± 1.12	5.33 ± 1.21	<0.01
Fibrinogen (g/L)	4.28 ± 1.15	4.44 ± 1.27	4.25 ± 1.07	4.39 ± 1.19	4.31 ± 1.13	4.47 ± 1.33	0.327
Ceruloplasmin (µmol/L)	1.04 ± 0.32	1.68 ± 0.57	1.92 ± 0.77	1.92 ± 0.22	2.39 ± 0.63	2.66 ± 0.12	<0.001
SBP (mmHg)	141 ± 3	159 ± 7	136 ± 3	150 ± 4	131 ± 4	147 ± 5	<0.01
DBP (mmHg)	72 ± 3	90 ± 2	69 ± 2	85 ± 2	67 ± 1	83 ± 2	0.178
S-Phosphorus (mmol/L)	1.20 ± 0.19	1.19 ± 0.11	1.14 ± 0.21	1.15 ± 0.16	1.11 ± 0.83	1.12 ± 0.71	0.256
U-Phosphorus (mmol/24 h)	302 ± 25	221 ± 17	248 ± 17	207 ± 21	222 ± 18	205 ± 55	<0.01

Metabolic Syndrome (MetS); CKD: Chronic Kidney Disease stage; SS: Southern Sahara; F: female; M: male; BMI: Body mass index; WC: Waist Circumference; WH: Waist Hips; BF: body fat percentage; HOMA: Homeostasis Model Assessment; C: cholesterol; HDL: high density lipoprotein; LDL: low density lipoprotein; SBP: systolic blood pressure; DBP: diastolic blood pressure; Hs-CRP: High sensitive C reactive Protein. The mean values are assigned from the standard error to the mean (X ± ESM). The degree of significance is calculated for a risk of error α = 5%. The mean comparison is established for each group, White SS and Black SS. The same baseline data obtained before vitamin D supplementation was used for both two groups: D2 (2000 IU) and D60 (60,000 IU). *p*-value is calculated at baseline time (White SS versus Black SS groups).

**Table 4 nutrients-14-02285-t004:** Effects of oral 2000 IU vitamin D3 supplementation/day at 24 and 36 weeks on serum vitamin D level, CKD clusters and thromboembolic biomarkers in Blacks versus Whites SS participants.

	Baseline	24 Weeks			36 Weeks		*p* Value
Parameters	WSS (*N* = 79)	BSS (*N* = 72)	WSS (*N* = 74)	BSS (*N* = 70)	WSS (*N* = 71)	BSS (*N* = 68)	
25(OH)D (ng/mL)	19.8 ± 4.33	6.98 ± 1.91	30.9 ± 5.18	19.1 ± 6.15	48.1 ± 4.22	26.8 ± 3.54	<0.001
1,25(OH)2D (pg/mL)	29.9 ± 7.51	44.3 ± 3.55	40.4 ± 1.66	72.6 ± 2.89	55.3 ± 3.11	87.9 ± 1.98	<0.001
eGFR (mL/minper1.73 m^2^)	48.2 ± 2.31	45.4 ± 2.29	52.2 ± 9.09	49.6 ± 7.11	61.6 ± 4.22	55.8 ± 3.11	<0.001
S-Creatinine (µmol/L)	168 ± 19	221 ± 27	146 ± 63	199 ± 18	130 ± 13	161 ± 22	<0.001
U-Creatinine (mmol/24 h)	10.9 ± 2.11	14.8 ± 2.17	13.7 ± 1.99	18.5 ± 3.34	17.6 ± 1.09	22.8 ± 1.77	<0.001
S-Creatinine-BMIratio	5.97 ± 1.22	6.84 ± 1.31	5.52 ± 1.11	6.25 ± 1.43	5.01 ± 1.31	5.99 ± 1.18	<0.001
S-Uricacid (µmol/L)	404 ± 36	441 ± 22	359 ± 17	410 ± 20	287 ± 32	366 ± 17	<0.001
U-Uricacid (mmol/24 h)	2.45 ± 0.66	2.71 ± 0.45	2.67 ± 0.19	2.97 ± 0.31	2.91 ± 0.24	3.21 ± 0.55	<0.001
S-Ca (mmol/L)	2.32 ± 0.62	2.20 ± 0.33	2.39 ± 0.61	2.44 ± 0.39	2.51 ± 0.16	2.65 ± 0.71	<0.02
U-Ca (mmol/24 h)	41.4 ± 3.41	25.2 ± 4.33	47.9 ± 6.11	29.7 ± 2.41	52.4 ± 1.99	32.4 ± 3.11	<0.02
iCa (mmol/24 h)	1.17 ± 0.15	1.30 ± 0.25	1.20 ± 0.19	1.41 ± 0.55	1.38 ± 0.11	1.49 ± 0.31	<0.02
S-Alb (µmol/L)	595 ± 44	580 ± 33	665 ± 71	609 ± 55	744 ± 24	720 ± 19	<0.001
U-Alb (mg/24 h)	40.5 ± 7.11	54.8 ± 3.27	31.9 ± 2.08	42.5 ± 1.44	29.7 ± 3.11	34.9 ± 2.71	<0.001
ApoproteinA_1_ (µmol/L)	42.4 ± 2.17	41.1 ± 3.66	44.1 ± 2.09	43.7 ± 3.11	46.9 ± 7.88	45.3 ± 3.77	<0.02
ApoproteinB_100_ (µmol/L)	35.9 ± 5.83	38.3 ± 7.15	33.4 ± 1.77	36.4 ± 4.51	32.9 ± 2.24	33.1 ± 1.91	<0.01
ApoB_100_-ApoA1ratio	0.846 ± 0.06	0.931 ± 0.07	0.757 ± 0.03	0.832 ± 0.05	0.702 ± 0.07	0.717 ± 0.04	<0.01
Lp(a) (nmol/L)	78.5 ± 4.10	94.5 ± 3.27	76.9 ± 6.11	92.8 ± 1.72	72.1 ± 2.66	91.2 ± 3.27	0.772
tHcy (µmol/L)	17.2 ± 2.52	21.1 ± 3.31	16.5 ± 2.15	19.8 ± 2.81	16.9 ± 3.55	20.7 ± 1.88	0.568

SS: Southern Sahara; eGFR: estimated Glomerular filtration rate; S: Serum; U: urinary; S-Ca: Serum Calcium; U-Ca: urinary calcium; iCa: ionized Calcium; S-Alb: Serum Albumin; U-Alb: Micro albuminuria; 25(OH)D: total serum 25 hydroxyvitamin D; 1,25(OH)2D: 1,25-dihydroxyvitamin D; Lp (a): lipoprotein (a); tHcy: total homocysteine. The mean values are assigned from the standard error to the mean (X ± ESM). The degree of significance is calculated for a risk of error α = 5%. The mean comparison is established for each group, White SS and Black SS. Baseline data were obtained before vitamin D supplementation. *p*-value is calculated at baseline time (White SS versus Black SS groups).

**Table 5 nutrients-14-02285-t005:** Effects of oral 60,000 IU vitamin D3 supplementation/month at 24 and 36 weeks on serum vitamin D level, CKD clusters and thromboembolic biomarkers in Blacks versus Whites SS participant’s.

	Baseline	24 Weeks			36 Weeks		*p* Value
Parameters	WSS (*N* = 79)	BSS (*N* = 72)	WSS (*N* = 74)	BSS (*N* = 70)	WSS (*N* = 71)	BSS (*N* = 68)	
25OHD (ng/mL)	19.8 ± 4.33	6.98 ± 1.91	31.5 ± 4.11	22.7 ± 5.09	46.3 ± 3.42	44.9 ± 2.45	<0.001
1,25(OH)2D (pg/mL)	29.9 ± 7.51	44.3 ± 3.55	41.2 ± 2.58	82.6 ± 3.77	56.2 ± 5.08	95.8 ± 2.09	<0.001
eGFR (mL/minper1.73 m^2^)	48.2 ± 2.31	45.4 ± 2.29	49.7 ± 5.32	55.1 ± 5.22	58.7 ± 4.11	61.9 ± 2.72	<0.001
S-Creatinine (µmol/L)	168 ± 19	221 ± 27	154 ± 44	178 ± 22	137 ± 25	144 ± 13	<0.001
U-Creatinine (mmol/L)	10.9 ± 2.11	14.8 ± 2.17	13.1 ± 2.08	20.5 ± 1.88	16.7 ± 3.11	25.1 ± 2.09	<0.001
S–Creatinine-BMIratio	5.97 ± 1.22	6.84 ± 1.71	5.25 ± 2.33	5.57 ± 2.07	4.76 ± 2.45	5.34 ± 1.87	<0.001
S-Uricacid (µmol/L)	404 ± 36	441 ± 22	360 ± 22	374 ± 31	289 ± 24	311 ± 55	<0.001
U-Uricacid (mmol/24 h)	2.45 ± 0.66	2.71 ± 0.45	2.62 ± 0.88	3.05 ± 0.14	2.95 ± 0.43	3.49 ± 0.27	<0.001
S-Ca (mmol/L)	2.32 ± 0.62	2.20 ± 0.33	2.40 ± 0.17	2.66 ± 0.88	2.55 ± 0.27	2.77 ± 0.19	<0.02
U-Ca (mmol/24 h)	41.4 ± 3.41	25.2 ± 4.33	48.7 ± 5.09	33.5 ± 1.18	53.5 ± 2.23	38.9 ± 3.51	<0.02
iCa (mmol/L)	1.17 ± 0.15	1.30 ± 0.25	1.19 ± 0.22	1.43 ± 0.19	1.39 ± 0.31	1.52 ± 0.15	<0.02
S-Alb (µmol/L)	595 ± 44	580 ± 33	632 ± 55	649 ± 23	747 ± 31	785 ± 27	<0.001
U-Alb(mg/24 h)	40.5 ± 7.11	54.8 ± 3.27	31.2 ± 1.31	37.8 ± 2.09	28.3 ± 4.07	31.4 ± 3.18	<0.001
ApoproteinA_1_ (µmol/L)	42.4 ± 2.17	41.1 ± 3.66	43.8 ± 1.98	44.9 ± 2.72	45.1 ± 5.09	47.2 ± 3.54	<0.001
ApoproteinB_100_ (µmol/L)	35.9 ± 5.83	38.3 ± 7.15	32.7 ± 1.33	34.3 ± 2.17	31.7 ± 1.89	30.1 ± 1.74	<0.02
ApoB_100_-ApoA1ratio	0.846 ± 0.06	0.931 ± 0.07	0.746 ± 0.02	0.763 ± 0.05	0.744 ± 0.05	0.637 ± 0.02	<0.01
Lp(a) (nmol/L)	78.5 ± 4.10	94.5 ± 3.27	77.5 ± 3.71	91.1 ± 1.22	76.9 ± 1.87	90.7 ± 2.37	0.401
tHcy (µmol/L)	17.2 ± 2.52	21.1 ± 3.31	16.1 ± 1.54	18.8 ± 1.27	15.8 ± 2.32	18.3 ± 1.09	0.711

SS: Southern Sahara; eGFR: estimated Glomerular filtration rate; S: Serum; U: urinary; S-Ca: Serum Calcium; U-Ca: urinary calcium; iCa: ionized Calcium; S-Alb: Serum Albumin; U-Alb: Micro albuminuria; 25(OH)D: total serum 25 hydroxyvitamin D; 1,25(OH)2D: 1,25-dihydroxyvitamin D; Lp (a): lipoprotein (a); tHcy: total homocysteine. The mean values are assigned from the standard error to the mean (X ± ESM). The degree of significance is calculated for a risk of error α = 5%. The mean comparison is established for each group, White SS and Black SS. Baseline data were obtained before vitamin D supplementation. *p*-value is calculated at baseline time (White SS versus Black SS groups).

## Data Availability

The data presented in this study are available on request from the corresponding author.

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
