# Peer review of "Oral Cholecalciferol Supplementation in Sahara Black People with Chronic Kidney Disease Modulates Cytokine Storm, Oxidative Stress Damage and Athero-Thromboembolic Risk"

_nutrients, 2022, doi:10.3390/nu14112285_

Round 1
Reviewer 1 Report
This randomized trial by Zoubiri and colleagues examined the treatment effects of cholecalciferol supplementation on biomarkers of immune, oxidative stress, and thromboembolic events according to race (Black versus White patients in Saharan Africa). The trial was generally well designed and executed (including randomization and treatment allocation, identification of the relevant study population and informed consent, selection of endpoints and assays for biomarkers) and it investigated a clinically important question. However, the presentation of the trial results is unfortunately too descriptive, as noted in my remarks below.
Title
- please consider shortening the title. Also, it is preferred to state race as "Black" instead of "Black-skinned." Please also note that race is capitalized.
Abstract
- the results lack detail on treatment effect sizes. What were the effect estimates and statistical tests for your hypotheses? These are provided in the body of the manuscript but it would be helpful to see them in the abstract as well.
Statistical Analysis
- the analysis is very descriptive, with no multivariable models to estimate treatment effect sizes. This is a substantial weakness in the reporting of the trial results. In this regard, the authors' conclusions about the study findings are not adequately supported by the data. The conclusions are primarily based on correlations and percentage change in biomarkers.
Discussion
- some of the citations referenced by the authors seem to be used out of context. For example, reference #33, DePasquale et al, 2020 does not report on vitamin D effects by race, as suggested by the authors. The reviewer did not check all the other references cited. Please ensure the appropriate citations are used to support the statements/conclusions drawn from the data.
Author Response
Point-by-point rebuttals / answers to the reviewer’s comments
Manuscript ID: Nutrients-1716500- Major Revisions
Date of submission: 22/04/2022 - Date of the reply to the submission: 16/05/2022 by the journal
Revised file within 10 days (26/05/2022)
Reviewer # 1
____________________________________
Title: Modulation of Chronic Kidney Disease in Black-skinned Sahara residents by Oral Cholecalciferol Supplementation: Interactions between Cytokines Storm, Oxidative Stress Damage and Athero-thromboembolic Risk
- Zoubiri1, 2, A. Tahar1, S. AitAbderrhmane 3, M. Saidani 4, E.A. Koceir1*
1 Bioenergetics and Intermediary Metabolism team, USTHB, Algiers, Algeria;
2 Biology and Physiology Laboratory, Algiers Algeria;
3 Diabetology unit, Seghir Nekkache Hospital, Algiers, Algeria;
4 Clinical Nephrology Exploration Unit, Dialysis and Kidney Transplantation Unit, University Hospital Center, Algiers,
* Corresponding author
General Comments
. English language and style: Extensive editing of English language and style required
- Does the introduction provide sufficient background and include all relevant references? Can be improved
- Are all the cited references relevant to the research? Must be improved
- Is the research design appropriate? Yes
- Are the methods adequately described? Yes
- Are the results clearly presented? Must be improved
- Are the conclusions supported by the results? Can be improved
Comments and Suggestions for Authors
This randomized trial by Zoubiri and colleagues examined the treatment effects of cholecalciferol supplementation on biomarkers of immune, oxidative stress, and thromboembolic events according to race (Black versus White patients in Saharan Africa). The trial was generally well designed and executed (including randomization and treatment allocation, identification of the relevant study population and informed consent, selection of endpoints and assays for biomarkers) and it investigated a clinically important question. However, the presentation of the trial results is unfortunately too descriptive, as noted in my remarks below.
- COMMENT: Title: please consider shortening the title. Also, it is preferred to state race as "Black" instead of "Black-skinned." Please also note that race is capitalized.
- ANSWER: We appreciate and fully agree with this comment. As suggested, we corrected the title. Please see the Track changes MS. We have deleted the words that have the same meaning and are not easily identifiable in the study, such as: Skinned, modulation, residents, and …interactions between...
Please see the new title in Track changes Manuscript (MS).
- COMMENT: Abstract: the results lack detail on treatment effect sizes. What were the effect estimates and statistical tests for your hypotheses? These are provided in the body of the manuscript but it would be helpful to see them in the abstract as well.
- ANSWER: We apologize for this unintentional omission for not specifying this methodological detail in the material section, because the instructions authors limit the words number in the Abstract. Please see the Track changes MS.
- COMMENT: Statistical Analysis: the analysis is very descriptive, with no multivariable models to estimate treatment effect sizes. This is a substantial weakness in the reporting of the trial results. In this regard, the authors' conclusions about the study findings are not adequately supported by the data. The conclusions are primarily based on correlations and percentage change in biomarkers.
- ANSWER: I agree. We appreciate this concern related to statistical analysis. We are very sorry to have omitted to explain it in the section material and methods, because we thought that it was not necessary to mention it, because the software treats the model regression. Please see the revised Materiel & methods section in Track changes MS.
- COMMENT: Discussion: some of the citations referenced by the authors seem to be used out of context. For example, reference #33, DePasquale et al, 2020 does not report on vitamin D effects by race, as suggested by the authors. The reviewer did not check all the other references cited. Please ensure the appropriate citations are used to support the statements/conclusions drawn from the data.
- ANSWER: We apologize for this reference #33 which non-intentional included. I want you to be convinced that all references used in this study are to support the statements/conclusions drawn from our data. However, there are 3 types of references: 1) Black race – CKD relationship; 2): vitamin D deficiency - Black race relationship; 3): and Black race - CKD - vitamin D deficiency relationship.
The Reference number 33 related to the prevalence of kidney failure in Black family for medical follow-up. We have replaced reference 33 with another recent study from 2019 by the same authors that discusses the Black race - CKD - vitamin D deficiency relationships. Please see the revised References section in Track changes MS.
New Ref. 33:
Lunyera J, Davenport CA, Pendergast J, Musani SK, Bhavsar NA, Sims M, Mwasongwe S, Wolf M, Diamantidis CJ, Boulware LE, Scialla JJ. Modifiers of Plasma 25-Hydroxyvitamin D and Chronic Kidney Disease Outcomes in Black Americans: The Jackson Heart Study. J Clin Endocrinol Metab. 2019 Jun 1; 104(6):2267-2276.

Reviewer 2 Report
The authors in this study tested the supplementation dose-response of 87 25-hydroxyvitamin D (Cholecalciferol) in continuously, low multi-doses (2,000 IU / day) 88 versus an intermittent, high dose (60,000 IU / month) in CKD 3 adult’s black versus white vitamin D deficiency participants from the same region of south-eastern Algeria. The objective of this investigation was to know whether the Vitamin D3 supplementation could modulate: the change in serum Vitamin D3 levels; the change in pro-inflammatory cytokines production, the oxidative stress damage, and atherogenicity risk.
The authors attribute all the beneficial changes observed during the therapy to the effect of cholecalciferol. However, the results describing both groups of people show important changes in BMI, WC, BF% e.g in WSS group BF% was from 46.6 to 39.3, and in BSS group BF% was from 55.1 to 53.9 . The authors should provide absolute values for body weight, fat mass, and fat-free mass for both groups in all phases. As is known, both in the general population and in people with CKD, the fat mass reduction produces many beneficial metabolic effects e.g. Kozlowska, 2010. Thus, it is impossible to assess whether the observed effects are the result of supplementation, loss of fat mass, or both. It is, therefore, necessary to change the title and many statements in the results, discussion, and conclusions e.g. “Based our clinical data, it is confirmed that cholecalciferol supplementation (25OHD-S) modulates the cytokine storm, oxidative stress damage and metabolic disorders. According to the racial-ethnic factor, it appears that 25OHD-S provides a significant 518 benefit both in black and white SS participants by stabilizing the renal disease and preventing its progression to end-stage renal disease.”
Lucyna Kozłowska, Andrzej Rydzewski, Bartosz Fiderkiewicz, Anna Wasińska-Krawczyk, Agnieszka Grzechnik, Danuta Rosołowska-Huszcz. Adiponectin, resistin and leptin response to dietary intervention in diabetic nephropathy. J Ren Nutr. 2010 Jul;20(4):255-62. doi: 10.1053/j.jrn.2010.01.009. Epub 2010 May 26.
The authors included the results of many analyzed biochemical parameters, but there are no basic parameters such as the concentration of calcium and phosphorus in the blood serum. As is known, cholecalciferol also increases the absorption of phosphorus (Nguyen, 2021), which is very dangerous in terms of hyperphosphatemia. This aspect should be included as research limitations.
Todd Nguyen, Deanna Joe, and Ankur D. Shah. Forget the phosphorus: A case of hypervitaminosis D-induced symptomatic hypercalcemia. Clinical Nephrology – Case Studies, Vol. 9/2021 (1-3)
An additional difficult aspect is the huge caloric intake in Blacks – 62 kcal/kg/day – this is impossible, this level of caloric intake is in elite athletes with high intense training periods. In comparison in Whites, caloric intake is 35 kcal/kg/day. Moreover, Blacks had much lower physical activity and yet had a significant reduction in BMI, BF%. This is inconsistent and requires verification.
Author Response
Point-by-point rebuttals / answers to the reviewer’s comments
Manuscript ID: Nutrients- 1716500- Major Revisions
Date of submission: 22/04/2022 - Date of the reply to the submission: 16/05/2022 by the journal
Revised file within 10 days (26/05/2022)
Reviewer # 2
____________________________________
Title: Modulation of Chronic Kidney Disease in Black-skinned Sahara residents by Oral Cholecalciferol Supplementation: Interactions between Cytokines Storm, Oxidative Stress Damage and Athero-thromboembolic Risk
- Zoubiri1, 2, A. Tahar1, S. Ait Abderrhmane 3, M. Saidani 4, E.A. Koceir1*
1 Bioenergetics and Intermediary Metabolism team, USTHB, Algiers, Algeria;
2 Biology and Physiology Laboratory, Algiers Algeria;
3 Diabetology unit, Seghir Nekkache Hospital, Algiers, Algeria;
4 Clinical Nephrology Exploration Unit, Dialysis and Kidney Transplantation Unit, University Hospital Center, Algiers, Algeria
* Corresponding author
General Comments
. English language and style: I don't feel qualified to judge about the English language and style
- Does the introduction provide sufficient background and include all relevant references? Can be improved
- Are all the cited references relevant to the research? Can be improved
- Is the research design appropriate? Must be improved
- Are the methods adequately described? Can be improved
- Are the results clearly presented? Can be improved
- Are the conclusions supported by the results? Must be improved
Comments and Suggestions for Authors
The authors in this study tested the supplementation dose-response of 25-hydroxyvitamin D (Cholecalciferol) in continuously, low multi-doses (2,000 IU / day) versus an intermittent, high dose (60,000 IU / month) in CKD 3 adult’s black versus white vitamin D deficiency participants from the same region of south-eastern Algeria. The objective of this investigation was to know whether the Vitamin D3 supplementation could modulate: the change in serum Vitamin D3 levels; the change in pro-inflammatory cytokines production, the oxidative stress damage, and atherogenicity risk.
- COMMENT: The authors attribute all the beneficial changes observed during the therapy to the effect of cholecalciferol. However, the results describing both groups of people show important changesin BMI, WC, BF% e.g in WSS group BF% was from 46.6 to 39.3, and in BSS group BF% was from 55.1 to 53.9.
- ANSWER: Thank you for this constructive comment. We apologize for not having detailed in the material & methods section the methodology that allowed the anthropometric results, because we should respect the word number required by the Nutrients journal in the instructions to authors. As suggested by the reviewer, we have added the method of feeding evaluation pattern and physical activity levels. Please see material & methods section (2.5. Metabolic syndrome clusters screening) in the revised MS. We agree with the reviewer that other factors could influence the CKD evolution in this study specific to ethnicity and race factors. The only common factor between the two ethnic groups was vitamin D supplementation tested at 2 different doses and 2 different durations. It is true that the dietary and physical activity factord are parameters influencing the the CKD evolution, but we could not argue all the parameters. Moreover, we studied the renin-angiotensin-aldosterone system, but we could not include it otherwise the article would be too long according to the instructions authors. In this study we focused on oxidative stress and inflammation according to the journal Nutrients special issue. However, we argued the effects of vitamin D on anthropometric parameters (BMI, WC and %fat mass) in discussion section.
- COMMENT: The authors should provide absolute values for body weight, fat mass, and fat-free mass for both groups in all phases. As is known, both in the general population and in people with CKD, the fat mass reduction produces many beneficial metabolic effects e.g. Kozlowska, 2010.
- ANSWER: I agree. Thank you very much for this critical comment. In this study, the fat mass was estimated by calculation method according to the Deurenberg formula (see material & methods section). Fat-free mass or muscle mass was not assessed in this study, because this anthropometric parameter is not included among our objectives. To do this, it was necessary to use the impedancemetry technique. For body weight (BW), several studies do not take account BW in monitoring vitamin D treatment, but rather BMI. It is for this reason that we have not included BW among parameters in the tables (2 and 3). We considered the interesting publication of Kozlowska, 2010 for discussion section in the revised MS.
- COMMENT: The authors included the results of many analyzed biochemical parameters, but there are no basic parameters such as the concentration of calcium and phosphorus in the blood serum. As is known, cholecalciferol also increases the absorption of phosphorus (Nguyen, 2021), which is very dangerous in terms of hyperphosphatemia. This aspect should be included as research limitations.
- ANSWER: I agree. Thank you very much for this pertinent comment. Data on serum calcium concentrations exist (S-Ca); they are included in Tables 4 and 5. Serum phosphorus concentrations were performed. We have included them in Tables 2 and 3. We have presented them in the results section and discussed in relation to the publication of Nguyen, 2021. Please see the revised MS.
- COMMENT: Thus, it is impossible to assess whether the observed effects are the result of supplementation, loss of fat mass, or both. It is, therefore, necessary to change the title and many statements in the results, discussion, and conclusions e.g. “Based our clinical data, it is confirmed that cholecalciferol supplementation (25OHD-S) modulates the cytokine storm, oxidative stress damage and metabolic disorders. According to the racial-ethnic factor, it appears that 25OHD-S provides a significant benefit both in black and white SS participants by stabilizing the renal disease and preventing its progression to end-stage renal disease.”
- ANSWER: I agree. Thank you very much for this very relevant comment. We have changed the title of the article. It is as follows: Effects of oral Cholecalciferol Supplementation on Cytokines Storm, Oxidative Stress Damage and Athero-thromboembolic Risk in Sahara Black with Chronic Kidney Disease.
- COMMENT: An additional difficult aspect is the huge caloric intake in Blacks – 62 kcal/kg/day – this is impossible, this level of caloric intake is in elite athletes with high intense training periods. In comparison in Whites, caloric intake is 35 kcal/kg/day. Moreover, Blacks had much lower physical activity and yet had a significant reduction in BMI, BF%. This is inconsistent and requires verification.
- ANSWER: Thank you very much for this very relevant comment. Our results are in agreement with several studies carried in the Black African-American subject where caloric intake value reaches 94.7 kcal linked to a lower resting metabolic rate in Black people compared to White people (https://pubmed.ncbi.nlm.nih.gov/22668852/; https://pubmed.ncbi.nlm.nih.gov/10393133/) which explains the prevalence of obesity in the black population. Regarding the effects of vitamin D supplementation on a significant BF% reduction, while the lower physical activity. This can be explained via the effects of leptin which promotes the reduction of caloric intake and increases basal energy expenditure. We are in agreement with several studies on this subject carried in the Black African-American subject. However, the data are controversial about vitamin D effect and BMI relationship. In the Black subject requires high doses of vitamin D to have physiological benefits compared to the White subject who requires low doses of vitamin D.

Reviewer 3 Report
The work carried out by Zoubiri et al. shows that vitamin D3 supplementation exerts a particularly important nephroprotective effect in black-skinned patients through anti-inflammatory and antioxidant mechanisms. It is a very complete paper, with an extensive introduction, a large number of results and a very good discussion. However, I consider that some errors should be corrected before its final publication:
- There are numerous errors in the writing of the article with respect to English. All text should be revised in its entirety to correct these errors. Here are some examples (there are more than these):
Line 31: "and the treatment duration doesn’t similar in black-skinned"...a verb is missing.
Line 32: "it appear"...should be appears.
Line 42: "including black and whites people"...should be white
Lines 175-176: "Vitamin D status was measured as serum 25(OH) D concentrations (including both D2 and D3 isomers) was performed using a high performance"...a connector is missing before the second was.
Line 516: "Based our clinical data"...should be based on
Line 520: "this investigation reveal that"...should be reveals.
Line 531: "melanin and a decrease dermal"...should be decreased
- There are other errors of form in the writing of the text:
Line 16: "chronic kidney disease" should not be capitalized.
Line 22: The abbreviation "DS3" is not previously described.
Line 30: "in the 02 groups"...why 02?
Line 45: "Chronic Kidney Disease (CKD) more than 10%...", only CKD can now be used, since the abbreviation has already been described previously.
Figures should be introduced in the text, so that the reader knows when he/she should look at them.
Figure 1: "White and Black SS CKD Participants 1, 2, 3, 4 (n=760 accepted to participate)"...What does "1, 2, 3, 4" mean?
Line 150: "Hs-CRP" ...this abbreviation has not yet been described.
Line 174: "Serum 25-OH Vitamin D and 1, 25(OH) 2D assessment"...what is 2D?
- Line 102: "The Whites (W) and Blacks (B) participants aged 40 to 60 years, according to skin color including white and black SS participants were recruited from 2015 to 2020"... information is repeated and also the phrase is not well understood.
- Lines 112-114: The dosage schedule is not well understood. This sentence should be corrected and explained better.
- Figure 1: "CKD with eGFR > 90 mL/min/1.73m2" appears as an exclusion criterion... this is not a type of CKD, if a patient has > 90 mL/min/1.73 m2 then they have the normal kidney function.
- Section 2.5: the methods used must be specifically detailed, indicating references of the commercial kits used and if commercial kits were not used, describing the complete method or including a bibliographic reference where it is described.
- Section 2.6. much of the information that appears here has already been described in section 2.5. Avoid repeating information.
- I think section 2.10 should come before section 2.1.
- In general, all the figures should be improved, as the letters and numbers are very small.
- Why has the statistical analysis that compares the efficacy of the two dosage regimens within the same type of population based on their skin color not been carried out? I believe that important results would also be obtained.
Author Response
Point-by-point rebuttals / answers to the reviewer’s comments
Manuscript ID: Nutrients- 1716500- Major Revisions
Date of submission: 22/04/2022 - Date of the reply to the submission: 16/05/2022 by the journal
Revised file within 10 days (26/05/2022)
Reviewer # 3
____________________________________
Title: Modulation of Chronic Kidney Disease in Black-skinned Sahara residents by Oral Cholecalciferol Supplementation: Interactions between Cytokines Storm, Oxidative Stress Damage and Athero-thromboembolic Risk
- Zoubiri1, 2, A. Tahar1, S. Ait Abderrhmane 3, M. Saidani 4, E.A. Koceir1*
1 Bioenergetics and Intermediary Metabolism team, USTHB, Algiers, Algeria;
2 Biology and Physiology Laboratory, Algiers Algeria;
3 Diabetology unit, Seghir Nekkache Hospital, Algiers, Algeria;
4 Clinical Nephrology Exploration Unit, Dialysis and Kidney Transplantation Unit, University Hospital Center, Algiers, Algeria
* Corresponding author
General Comments
. English language and style: Moderate English changes required.
- Does the introduction provide sufficient background and include all relevant references? Yes
- Are all the cited references relevant to the research? Yes
- Is the research design appropriate? Yes
- Are the methods adequately described? Can be improved
- Are the results clearly presented? Can be improved
- Are the conclusions supported by the results? Yes
Comments and Suggestions for Authors
The work carried out by Zoubiri et al. shows that vitamin D3 supplementation exerts a particularly important nephroprotective effect in black-skinned patients through anti-inflammatory and antioxidant mechanisms. It is a very complete paper, with an extensive introduction, a large number of results and a very good discussion. However, I consider that some errors should be corrected before its final publication: - There are numerous errors in the writing of the article with respect to English. All text should be revised in its entirety to correct these errors. Here are some examples (there are more than these):
We do appreciate and fully agree with this comment. The revised version of the Manuscript (MS) has been checked duly and thoroughly for the grammatical and typographical mistakes. We have revised our MS by a native English speaker. However, if the MS is accepted for publication and the reviewers decides to correct the MS by the MDPI editing English correction system (Regular grammar check), we agree to pay the correction fee.
NOTE: Words and phrases highlighted in pink color are deleted in new Track changes MS.
- COMMENT: Line 31: "and the treatment duration doesn’t similar in black-skinned"...a verb is missing.
- ANSWER: Thank you for this constructive comment. We wrote the sentence without changing the meaning of the text, as follows: ‘We have shown that the dose and duration of 25OHD-S treatment are not similar in Black SS residents compared to white SS subjects’
Please see the revised Abstract-MS.
- COMMENT: Line 32: "it appear"...should be appears.
- ANSWER: I agree. Thank you very much for this correction. Please see the revised Abstract-MS.
- COMMENT: Line 42: "including black and whites people"...should be white
- ANSWER: I agree. Thank you very much for this correction. Please see the revised Abstract-MS.
- COMMENT: Lines 175-176: "Vitamin D status was measured as serum 25(OH) D concentrations (including both D2 and D3 isomers) was performed using a high performance"...a connector is missing before the second was.
- ANSWER: I agree. Thank you very much for this very relevant comment. This specific part to the vitamin D assay protocol has been rewritten and detailed. Please see the revised Materiel & Methods section-MS.
- COMMENT: Line 516: "Based our clinical data"...should be based on
- ANSWER: I agree. Thank you very much for this correction. Please see the revised MS.
- COMMENT: Line 520: "this investigation reveal that"...should be reveals.
- ANSWER: I agree. Thank you very much for this correction. Please see the revised MS.
- COMMENT: Line 531: "melanin and a decrease dermal"...should be decreased
- ANSWER: I agree. Thank you very much for this correction. Please see the revised MS.
- COMMENT: Line 16: "chronic kidney disease" should not be capitalized.
- ANSWER: I agree. Thank you for this important clarification. Please see the revised Abstract-MS.
- COMMENT: Line 22: The abbreviation "DS3" is not previously described.
- ANSWER: I agree. Thank you very much for this attention. DS3 has been removed. In the first version of the MS, we chose as abbreviation for vitamin D supplementation: SD3; but later we changed to 25OHD-S. Please see the revised Abstract-MS.
- COMMENT: Line 30: "in the 02 groups"...why 02?
- ANSWER: I agree. Thank you very much for this remark. Vitamin D supplementation is beneficial in the 02 groups (Black and White people) link to remove the deficiency state (> 30 ng / ml plasma); however the supplementation protocol is different in the Black subject compared to the White subject. In the Black it is necessary to high dose at 60,000 IU/month and to maintain the treatment for 36 weeks (intermittent), whereas in the white subject, a low dose of 2,000 IU/day and a continuously treatment of 24 weeks are sufficient.
- COMMENT: Line 45: "Chronic Kidney Disease (CKD) more than 10%...", only CKD can now be used, since the abbreviation has already been described previously.
- ANSWER: I agree. Thank you very much for this comment. You are right. Please see the revised MS.
- COMMENT: Figures should be introduced in the text, so that the reader knows when he/she should look at them.
- ANSWER: I agree. Thank you very much for this observation. Figures are already introduced in the results section, however as suggested by the reviewer, Figures should be reintroduced a second time in the discussion section. Please see the revised MS.
- COMMENT: Figure 1: "White and Black SS CKD Participants 1, 2, 3, 4 (n=760 accepted to participate)"...What does "1, 2, 3, 4" mean?
- ANSWER: Thank you for this comment. The numbers 1, 2, 3, 4 represent the four stages of CKD on the basis of eGFR. Please see the revised Figure 1 legend. For more details attached the link (https://www.kidneyfund.org/all-about-kidneys/stages-kidney-disease) or reference: https://pubmed.ncbi.nlm.nih.gov/21840587/. Stage 1 with normal or high eGFR (GFR > 90 mL/min), Stage 2 with mild CKD (GFR = 60-89 mL/min), Stage 3A with moderate CKD (GFR = 45-59 mL/min), Stage 3B with moderate CKD (GFR = 30-44 mL/min), Stage 4 with severe CKD (GFR = 15-29 mL/min) and Stage 5 with moderate CKD (GFR <15 mL/min).
- COMMENT: Line 150: "Hs-CRP" ...this abbreviation has not yet been described.
- ANSWER: We apologize do not clarify the abbreviation of this inflammatory parameter: Hs-CRP (high-sensitive C-reactive protein). Hs-CRP is the biomarker for detecting low-grade inflammation. Please see material & method section in the revised MS.
- COMMENT: Line 174: "Serum 25-OH Vitamin D and 1, 25(OH) 2D assessment"...what is 2D?
- ANSWER: We apologize do not clarify 2D. This means a double hydroxylation of vitamin D: the first is located in the liver (1D) to form the 25-OH Vitamin D and the second is located in the kidney (2D) to synthesize 1, 25(OH) 2D. Please see the reference: https://pubmed.ncbi.nlm.nih.gov/25496802/. Please see material & method section in the revised MS.
- COMMENT: - Line 102: "The Whites (W) and Blacks (B) participants aged 40 to 60 years, according to skin color including white and black SS participants were recruited from 2015 to 2020"... information is repeated and also the phrase is not well understood.
- ANSWER: We apologize for this misunderstood sentence. We wrote this sentence: “White (W) and Black (B) participants were recruited into the study in 2015 to 2020; they are old 40 to 60 years”. Please see material & method section in the revised MS.
- COMMENT: Lines 112-114: The dosage schedule is not well understood. This sentence should be corrected and explained better.
- ANSWER: We apologize for this misunderstood dosage schedule. We wrote this clinical protocol. Please see material & method section in the revised MS.
- COMMENT: - Figure 1: "CKD with eGFR > 90 mL/min/1.73m2" appears as an exclusion criterion... this is not a type of CKD, if a patient has > 90 mL/min/1.73 m2then they have the normal kidney function.
- ANSWER: It's true. We apologize for not written without CKD in figure 1. Please see the revised MS.
- COMMENT: - Section 2.5: the methods used must be specifically detailed, indicating references of the commercial kits used and if commercial kits were not used, describing the complete method or including a bibliographic reference where it is described.
- ANSWER: Thank you for this comment. As suggested by the reviewer, this entire part of Section 2.5 has been detailed by the specific assay methods for each measured parameter including a bibliographic reference. We didn't detail it in the first version of the MS, because the number of words according to the instructions to the authors is limited. Please see material & method section in the revised MS.
- COMMENT: - I think section 2.10 should come before section 2.1.
- ANSWER: Thank you for this comment. As suggested by the reviewer, section 2.10 is incorporated into the text before section 2.1. Please see material & method section in the revised MS.
- COMMENT: - Section 2.6. much of the information that appears here has already been described in section 2.5. Avoid repeating information.
- ANSWER: Thank you for this comment. We did not attention that we repeated ourselves in the parameters cited before. We removed section 2.6. Please see material & method section in the revised MS.
- COMMENT: - In general, all the figures should be improved, as the letters and numbers are very small.
- ANSWER: Thank you for this comment. I think the figures have been shrunk by the MDPI system that built the MS pdf. If you take the MS word version, you can enlarge the figure and the letters and numbers become better-quality. Please see in the revised MS (word version).
- COMMENT: - Why has the statistical analysis that compares the efficacy of the two dosage regimens within the same type of population based on their skin color not been carried out? I believe that important results would also be obtained.
- ANSWER: That's right. Thank you for this comment and Thank you very much for this advice. It is part of our perspectives. The initial idea of this study was linked to the prevalence of arterial hypertension associated to renal disorder was more important in the Algerian Sahara Black population than in the Algerian Sahara White population. Vitamin D supplementation was introduced by nephrologists to improve anti-hypertensive therapy, because in Black subjects, a resistance to treatment is important compared to White subjects. Following Vitamin D supplementation we noticed that the renal function improved further (decrease creatinine and microalbuminuria), which led us to study the dose factor and duration of treatment.

Round 2
Reviewer 2 Report
- COMMENT: The authors attribute all the beneficial changes observed during the therapy to the effect of cholecalciferol. However, the results describing both groups of people show important changes in BMI, WC, BF% e.g in WSS group BF% was from 46.6 to 39.3, and in BSS group BF% was from 55.1 to 53.9.
ANSWER: Thank you for this constructive comment. We apologize for not having detailed in the material & methods section the methodology that allowed the anthropometric results, because we should respect the word number required by the Nutrients journal in the instructions to authors. As suggested by the reviewer, we have added the method of feeding evaluation pattern and physical activity levels. Please see material & methods section (2.5. Metabolic syndrome clusters screening) in the revised MS. We agree with the reviewer that other factors could influence the CKD evolution in this study specific to ethnicity and race factors. The only common factor between the two ethnic groups was vitamin D supplementation tested at 2 different doses and 2 different durations. It is true that the dietary and physical activity factord are parameters influencing the the CKD evolution, but we could not argue all the parameters. Moreover, we studied the renin-angiotensin-aldosterone system, but we could not include it otherwise the article would be too long according to the instructions authors. In this study we focused on oxidative stress and inflammation according to the journal Nutrients special issue. However, we argued the effects of vitamin D on anthropometric parameters (BMI, WC and %fat mass) in discussion section.
- COMMENT: The authors should provide absolute values for body weight, fat mass, and fat-free mass for both groups in all phases. As is known, both in the general population and in people with CKD, the fat mass reduction produces many beneficial metabolic effects e.g. Kozlowska, 2010.
ANSWER: I agree. Thank you very much for this critical comment. In this study, the fat mass was estimated by calculation method according to the Deurenberg formula (see material & methods section). Fat-free mass or muscle mass was not assessed in this study, because this anthropometric parameter is not included among our objectives. To do this, it was necessary to use the impedancemetry technique. For body weight (BW), several studies do not take account BW in monitoring vitamin D treatment, but rather BMI. It is for this reason that we have not included BW among parameters in the tables (2 and 3). We considered the interesting publication of Kozlowska, 2010 for discussion section in the revised MS.
Regarding the answers to the comments 1 and 2.
Please provide information on body weight (table 2 and 3). With such significant changes in BMI and BF%, the obtained effect cannot be attributed solely to supplementation - it is obvious and requires a change of title and conclusions.
- COMMENT: An additional difficult aspect is the huge caloric intake in Blacks – 62 kcal/kg/day – this is impossible, this level of caloric intake is in elite athletes with high intense training periods. In comparison in Whites, caloric intake is 35 kcal/kg/day. Moreover, Blacks had much lower physical activity and yet had a significant reduction in BMI, BF%. This is inconsistent and requires verification.
ANSWER: Thank you very much for this very relevant comment. Our results are in agreement with several studies carried in the Black African-American subject where caloric intake value reaches 94.7 kcal linked to a lower resting metabolic rate in Black people compared to White people (https://pubmed.ncbi.nlm.nih.gov/22668852/; https://pubmed.ncbi.nlm.nih.gov/10393133/) which explains the prevalence of obesity in the black population. Regarding the effects of vitamin D supplementation on a significant BF% reduction, while the lower physical activity. This can be explained via the effects of leptin which promotes the reduction of caloric intake and increases basal energy expenditure. We are in agreement with several studies on this subject carried in the Black African-American subject. However, the data are controversial about vitamin D effect and BMI relationship. In the Black subject requires high doses of vitamin D to have physiological benefits compared to the White subject who requires low doses of vitamin D.
Regarding the answer to the comment 3
Please check the caloric content of the diet in Black.
Please read the suggested publication carefully - the average caloric content of the diet in this study was about 28 kcal / kg / day. Please see what the value of 94.7 kcal applies to.
Author Response
Point-by-point rebuttals / answers to the reviewer’s comments
Manuscript ID: Nutrients- 1716500- Major Revisions
Date of submission: 22/04/2022 - Date of the reply to the submission: 16/05/2022 by the journal
Revised file within 10 days (26/05/2022)
Reviewer # 2
____________________________________
Round 2
Title: Modulation of Chronic Kidney Disease in Black-skinned Sahara residents by Oral Cholecalciferol Supplementation: Interactions between Cytokines Storm, Oxidative Stress Damage and Athero-thromboembolic Risk
- Zoubiri1, 2, A. Tahar1, S. Ait Abderrhmane 3, M. Saidani 4, E.A. Koceir1*
1 Bioenergetics and Intermediary Metabolism team, USTHB, Algiers, Algeria;
2 Biology and Physiology Laboratory, Algiers Algeria;
3 Diabetology unit, Seghir Nekkache Hospital, Algiers, Algeria;
4 Clinical Nephrology Exploration Unit, Dialysis and Kidney Transplantation Unit, University Hospital Center, Algiers, Algeria
* Corresponding author
General Comments
. English language and style: I don't feel qualified to judge about the English language and style
- Does the introduction provide sufficient background and include all relevant references? Can be improved
- Are all the cited references relevant to the research? Can be improved
- Is the research design appropriate? Can be improved
- Are the methods adequately described? Yes
- Are the results clearly presented? Can be improved
- Are the conclusions supported by the results? Can be improved
Comments and Suggestions for Authors
- COMMENT: Regarding the answers to the comments 1 and 2.
- Please provide information on body weight (table 2 and 3).
- With such significant changes in BMI and BF%, the obtained effect supplementation –
- It is obvious and requires a change of title and
- Conclusions.
- ANSWER: Thank you for this constructive comment.
- We apologize for not added the body weight (BW) in table 2 and 3. Please see the revised MS.
- The significant changes in BMI and BF%, cannot be attributed solely to vitamin D supplementation. Indeed, it seems that in Black subjects the association vitamin D - Calcium supplementation is more effective than vitamin D alone. As mentioned in Tables 3 and 4, the results can be explained by increases the ionized calcium (iCa) in the Black subject compared to the White people. Thus, the sequestration of vitamin D in the adipose tissue of Black subjects in the presence of calcium can activate lipolysis signaling pathways. Overall, there is consistent evidence that calcium and vitamin D increase whole body fat oxidation and the lipolytic effects of calcium are admitted in some studies. As several studies and other trials show that vitamin D supplementation may be significantly associated with less weight gain but that this association may be dependent on adjunctive calcium supplementation and a particular region of fat. Please see the revised MS.
- As suggested by the reviewer, we have changed the title and conclusions.
Previous title: Effects of oral Cholecalciferol Supplementation on Cytokines Storm, Oxidative Stress Damage and Athero-thromboembolic Risk in Sahara Black with Chronic Kidney Disease.
New title: Oral Cholecalciferol Supplementation in Sahara Black with Chronic Kidney Disease can improve Cytokines Storm, Oxidative Stress Damage and Athero-thromboembolic Risk
- Conclusions: We have suggested that vitamin D supplementation can contribute to the improvement of CKD evolution, but not replace medical therapy.
- COMMENT: Regarding the answer to the comment 3. Please check the caloric content of the diet in Black. Please read the suggested publication carefully - the average caloric content of the diet in this study was about 28 kcal / kg / day. Please see what the value of 94.7 kcal applies to.
- ANSWER: I agree. Thank you very much for this critical comment.
Regarding the publication number 103 cited as reference; Thank you to read in the abstract- results section: “Mean intake was 94.7 kcal higher in AA women than in non-AA women (P < .05)”
The Black subject (AA) consumes 94.7 kcalorie in more compared to the White subject (non-AA). The authors explain the weight gain and obesity in the Black subject by the lower resting metabolic rate (basal metabolism) which is reduced compared to the White subject. To avoid any confusion, we have removed the value 94.7 kcalories, but we have left the argument of rate basal metabolic which decrease in Black subject. Please see the revised MS.

Reviewer 3 Report
The article has improved remarkably. Only, I still think that there are quite a few writing errors in English, so I recommend a review by an expert to solve these problems.
Author Response
Point-by-point rebuttals / answers to the reviewer’s comments
Manuscript ID: Nutrients- 1716500- Major Revisions
Date of submission: 22/04/2022 - Date of the reply to the submission: 16/05/2022 by the journal
Revised file within 10 days (26/05/2022)
Reviewer # 3
____________________________________
Round 2
Title: Modulation of Chronic Kidney Disease in Black-skinned Sahara residents by Oral Cholecalciferol Supplementation: Interactions between Cytokines Storm, Oxidative Stress Damage and Athero-thromboembolic Risk
- Zoubiri1, 2, A. Tahar1, S. Ait Abderrhmane 3, M. Saidani 4, E.A. Koceir1*
1 Bioenergetics and Intermediary Metabolism team, USTHB, Algiers, Algeria;
2 Biology and Physiology Laboratory, Algiers Algeria;
3 Diabetology unit, Seghir Nekkache Hospital, Algiers, Algeria;
4 Clinical Nephrology Exploration Unit, Dialysis and Kidney Transplantation Unit, University Hospital Center, Algiers, Algeria
* Corresponding author
General Comments
. English language and style: Moderate English changes required.
- Does the introduction provide sufficient background and include all relevant references? Yes
- Are all the cited references relevant to the research? Yes
- Is the research design appropriate? Yes
- Are the methods adequately described? Yes
- Are the results clearly presented? Yes
- Are the conclusions supported by the results? Yes
Comments and Suggestions for Authors
The article has improved remarkably. Only, I still think that there are quite a few writing errors in English, so I recommend a review by an expert to solve these problems.
We do appreciate and fully agree with this comment. We would like to thank you very much for your encouragement. The re-revised Track-version of the Manuscript (MS) has been checked duly and thoroughly for the grammatical and typographical mistakes. We have revised our MS by a colleague Professor Teacher in English. However, if the MS is accepted for publication we agree to pay the correction fee to MDPI editing English correction system (Regular grammar check)
